# Tracking and Understanding Object Transformations

**Yihong Sun**
Cornell University

**Xinyu Yang**
Cornell University

**Jennifer J. Sun**
Cornell University

**Bharath Hariharan**
Cornell University

## Abstract

Real-world objects frequently undergo state transformations. From an apple being cut into pieces to a butterfly emerging from its cocoon, tracking through these changes is important for understanding real-world objects and dynamics. However, existing methods often lose track of the target object after transformation, due to significant changes in object appearance. To address this limitation, we introduce the task of Track Any State: tracking objects through transformations while detecting and describing state changes, accompanied by a new benchmark dataset, VOST-TAS. To tackle this problem, we present TubeletGraph, a zero-shot system that recovers missing objects after transformation and maps out how object states are evolving over time. TubeletGraph first identifies potentially overlooked tracks, and determines whether they should be integrated based on semantic and proximity priors. Then, it reasons about the added tracks and generates a state graph describing each observed transformation. TubeletGraph achieves state-of-the-art tracking performance under transformations, while demonstrating deeper understanding of object transformations and promising capabilities in temporal grounding and semantic reasoning for complex object transformations. Code, additional results, and the benchmark dataset are available at https://tubelet-graph.github.io.

## 1 Introduction

As the Greek philosopher Heraclitus noted, nothing is permanent but change. All around us, objects undergo transformations that can dramatically alter their appearance, geometry, and sometimes even their identities. In nature, seeds give birth to plants, chicks emerge from eggs and a caterpillar metamorphoses into a butterfly, while in our homes we slice apples and tomatoes, fold clothes and build up chairs from pieces of wood (Figure 1). Understanding and tracking these transformations is important for modern vision systems. For instance, embodied agents like kitchen robots need to understand object pre- and post-conditions (such as the locations of sliced pieces of apples) to ground actions [22]. As another example, wildlife monitoring systems must recognize and keep track of the the butterflies emerging out of their chrysalis to keep tabs on the insect population. More generally, understanding and tracking object transformations can improve capabilities in action-grounding [36, 50], video editing [24], and scene modeling for augmented reality [56].

With these motivations in mind, we seek a system that, given a video and a prompt specifying a particular object, maps out how the object evolves over time, detects state changes, and tracks the resulting objects of these changes. We call this problem **Track Any State**. We observe that this is a strictly harder problem than object tracking on the one hand (which does not care about object transformations) and recognizing state change on the other (which does not track the change pre- and post-conditions in space and time). Combining tracking with state change produces a more complete representation that is useful for downstream tasks (Figure 1, bottom).

However, even the simpler problem of tracking objects through state transformations is challenging for existing methods. Object trackers of all kinds (be they based on template matching [18], optical flow [40], or supervised neural networks [35]) rely primarily on objects appearance, assuming that they do not change drastically across time. However, as shown in Figure 1, transformations or state

39th Conference on Neural Information Processing Systems (NeurIPS 2025).

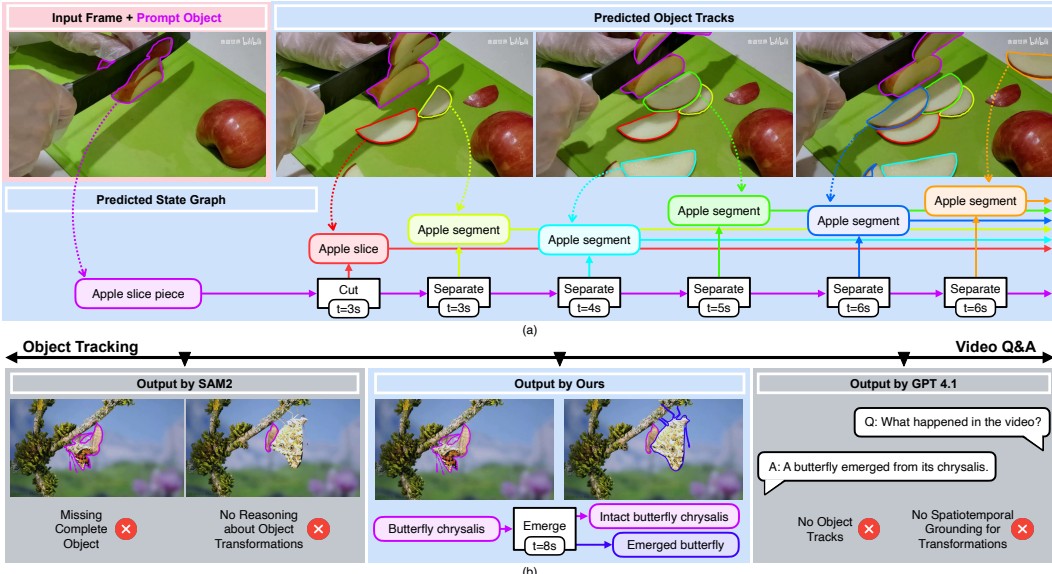

Figure 1: (*top*) Given a video and a object mask as prompt, TubeletGraph tracks the object consistently, while building a state graph for each detected transformation and its resulting effect. (*bottom*) Compared to existing object trackers (SAM2 [35]) or video Q&A systems (GPT-4 [1]), TubeletGraph predicts complete object tracks while providing spatiotemporal grounding for the transformation.

changes can alter object appearance significantly: e.g. from a red apple to a pile of white flesh pieces, or from a chrysalis to an empty chrysalis shell and a butterfly. These drastic changes cause existing trackers to fail in the face of these transformations, precluding any understanding of the state change.

Intriguingly, we find that the errors caused by tracking through state change are typically one-sided – when object appearance changes, the model is likely to predict the initial prompt object to be "missing", leading to false negatives. This observation offers an opportunity: if we can detect when these false negatives occur, we can attempt to both recover the missed object and understand the transformation that caused the error in the first place. To do this, we must answer two questions. First, when and where can we recover the missed object? In particular, how do we navigate the exponentially large search space among all pixels in the video to find the missing object? Second, how can we model the underlying transformations and resolve any object ambiguity after a state change? For instance, how can we name the transition and resulting objects in Figure 1 (bottom)?

We answer these two questions with TubeletGraph, a novel zero-shot framework for tracking and understanding object transformations in videos. First, to recover the missed object, we propose a new representation that dramatically reduces the search space. This representation tracks every entity in the video from the first frame and initiates new tracks in intermediate frames wherever there are untracked pixels. This produces a soup of "tubelets". Finding the missing post-condition object of a transformation then boils down to finding the right tubelet. By reasoning jointly about the semantics and spatial proximity of each tubelet, we demonstrate effective recovery of the missing object.

Second, we use the emergence of these new tubelets as a marker for when state transformations happen. We then query existing multi-modal LLMs [1] to describe the transformation and the resulting objects in natural language to produce a corresponding state graph. Together with the tracked tubelets, we can build a complete representation of the object's evolution over time. An example of TubeletGraph's output is shown in Figure 1 (top). In sum, our contributions are:

(1) We introduce Track Any State: a task of tracking objects through transformations while detecting and describing state changes, accompanied by VOST-TAS, a new benchmark dataset.

(2) We propose TubeletGraph: a zero-shot framework that recovers missing objects post-transformation by using a spatiotemporal partition of the video and constructs a state graph to detect and describe the underlying transformations.

(3) We demonstrate both state-of-the-art tracking performance under transformations as well as effective detection and description of the transformation itself.

## 2 Related Works

### 2.1 Object Tracking

**Benchmarks.** Object Tracking [37] aims to segment a target object in a given video. Similar to Semi-supervised Video Object Segmentation (VOS) [46, 31, 32], the target object is specified via a mask in the initial frame. In addition, recent benchmarks are proposed to address more challenging scenarios, including long videos [16, 21], crowds and occlusions [10, 13], and object motions [9, 13].

**Methods.** To predict consistent object tracks, prior works have mostly relied on appearance similarities via online feature finetuning [3, 4, 27], template matching [17, 43, 49], or attention-based memory reading [7, 28, 29, 51, 52, 6]. Recently, SAM2 [35] was proposed to extend Segment Anything Model (SAM) [19] for interactive video segmentation. By incorporating a memory-attention mechanisms, SAM2 enables object tracking by establishing consistent temporal object correspondences. SAM2Long [11] extends SAM2 and addresses error accumulation in long videos by maintaining multiple candidate tracks in a constrained tree search. Also, SAMURAI [48] introduces motion-based memory selections to handle crowded scenes with fast-moving or self-occluded objects. In addition, DAM4SAM [42] introduces a distractor-resolving memory to handle visually similar distractors.

While SAM2 and its variants demonstrate impressive results, they struggle when object appearance changes due to transformation. In our work, we first identify and reason about objects that are originally missed by SAM2 due to their transformations. Upon their successfully retrieval, TubeletGraph proceeds to leverage them as markers for event boundaries [55] to construct a state graph describing the transformations that cause the false negative errors as well as the recovered object themselves.

### 2.2 Understanding Object Transformations

Understanding object transformations in videos has been well-studied. VOST [41] and VSCOS [53] propose to focus on object transformations from human actions in ego-centric datasets [8, 14]. Similarly, M$^3$-VOS [5] extends the focus to objects undergoing phase (gas/liquid/solid) transitions. By assuming that object disorder increases through transformations, Re-VOS [5] propose to combine forward and reverse memory to improve object tracking through transformations.

Beyond object tracking, DTTO [45] provides box-level annotations for transforming objects while HowToChange [47] focuses on open-world localization of three stages (initial, transitioning, and end states) of object transformation. Also, WhereToChange [26] annotates spatially-progressing object state changes with the actionable and transformed object regions. For HowToChange [47] and WhereToChange [26], pseudo-labels are generated from off-the-shelf vision-language systems to train a video model for the respective task. Building upon the spatially-progressing state change segmentation maps, SPARTA [25] demonstrates real-world robotic manipulation capabilities such as spreading, mashing, and slicing.

In comparison, we focus on Track Any State, simultaneously tracking objects through transformations and detecting and naming the transformation.

### 2.3 Vision and Language

Recently, multi-modal systems have been proposed to integrate vision and language to understand and predict across modalities. CLIP [34] learns visual concepts from natural language captions via contrastive learning. From a shared embedding space for image and text, it enables zero-shot transfer to downstream tasks. FC-CLIP [54] further demonstrates this capability by predicting open-vocabulary segmentation using a frozen CLIP backbones. Finally, multi-modal LLMs such as GPT-4 [1] and Gemini [39] can reason about the visual/textual queries and generate natural language responses to further aid down-stream tasks [23, 12, 44, 15, 38].

In our work, we leverage CLIP to semantically reason about candidate spatiotemporal tubelets. Furthermore, by prompting GPT-4 [1] with the retrieved candidates, TubeletGraph constructs a state graph by parsing the description of the transformation and the resulting transformed objects.

# 3 Method

## 3.1 Task Formulation

In this paper, we propose the problem of Track Any State: tracking objects through transformations while detecting and naming the transformation.

Concretely, the input is a video $\mathcal{V} = \{I_t\}$ and a binary mask $\mathcal{M}_1$ in frame $I_1$ as the initial object prompt. The output is two-fold:

(1) A collection of **tracks** $\mathcal{T} = \{T^1, \dots T^n\}$, where each track $T^i$ corresponds to a mask $\mathcal{M}_t^i$ at each time step $t$. We allow for a collection of tracks rather than a single track because when objects undergo state change, they may break up into multiple independent parts; all of these are supposed to be tracked. Thus, $\mathcal{T}$ should track all segments that were created from the original object.

(2) A collection of **state changes** $\mathcal{S}$ where each state change $s \in \mathcal{S}$ is represented by a tuple $(t, \mathcal{T}_{\text{pre}}, \mathcal{T}_{\text{post}}, D)$. Here $t$ is the time step where the change happened, $\mathcal{T}_{\text{pre}}$ is the set of tracks involved before the change, $\mathcal{T}_{\text{post}}$ is the set of tracks involved after the change and $D$ is a description of the change.

This output can be visualized as in Figure 1, where each track in $\mathcal{T}$ is visualized as masks of a specific color, and the set of state changes $\mathcal{S}$ are visualized as a graph over the color-coded tracks.

**Overview of approach:** We now describe our approach, which we call **TubeletGraph**. Briefly, our approach first partitions the video into a set of tubelets $\mathcal{P}$, which are partial tracks by SAM2 and as such are delimited by appearance changes (Figure 2, top) . We then use notions of spatial and semantic proximity to the initial object prompt to decide which tracks to include in $\mathcal{T}$ (Figure 2, middle-left). For each track that gets added, we prompt a vision-language model to name the state change and the pre- and post-effects (Figure 2, middle-right). The end result is consistent object tracks through transformation and a state graph that describes the underlying transformation and resulting objects in natural language (Figure 2, bottom).

We now describe each step of this pipeline in detail.

## 3.2 Partitioning the Video into Tubelets

When objects undergo transformations, existing methods like SAM2 [35] often fail because (1) appearance information is no longer reliable when the object transforms, and (2) the assumption that the target object remains as a singular connected component no longer holds when the object fragments or decomposes.

As a result, these limitations often manifest in false negative errors. In the example of "taking a sheet of foil out of the foil box" (Figure 2), the appearance and geometry of the foil box object can change drastically when a sheet of foil separates from the box. If we only track the foil box (denoted in a pink contour) from the first frame, the foil sheet (an additional connected component with minimal appearance similarity) will be ignored in later frames.

To retrieve missing tracks like this and capture the full transformation of the object, we construct a spatiotemporal partition of the video $\mathcal{P}$ (Figure 2, top) to drastically reduce the search space. We first adopt an entity segmentation model, CropFormer (CF) [33], to obtain a complete spatial partition $\mathcal{E}_1$ of the initial frame $I_1$.

$$\mathcal{E}_1 = \text{CF}(I_1) \cup \{\mathcal{M}_1\} \tag{1}$$

where $\mathcal{E}_1$ represents the set of entity masks (including the object prompt $\mathcal{M}_1$) in frame $I_1$.[1] Then, we track each entity $e_1^i \in \mathcal{E}_1$ via SAM2 [35] to obtain a pool of tubelets $\mathcal{P}_{\text{init}} = \{P_i\}_{i=1}^{|\mathcal{E}_1|}$, where each tubelet $P_i = \{e_t^i\}_{t=1}^T$ represents the evolution of entity $e_1^i \in \mathcal{E}_1$ in the given video.

As one temporally progresses in the video, there will naturally be track-less regions where none of the initial tracked entities in $\mathcal{P}_{\text{init}}$ are present. Thus $\mathcal{P}_{\text{init}}$ is an incomplete spatiotemporal partition of the video. To complete this partition, we initialize the spatiotemporal partition $\mathcal{P}$ with $\mathcal{P}_{\text{init}}$ and

---

[1]Please refer to Appendix A.3 on resolving overlaps between the automatically segmented entities and $\mathcal{M}_1$.

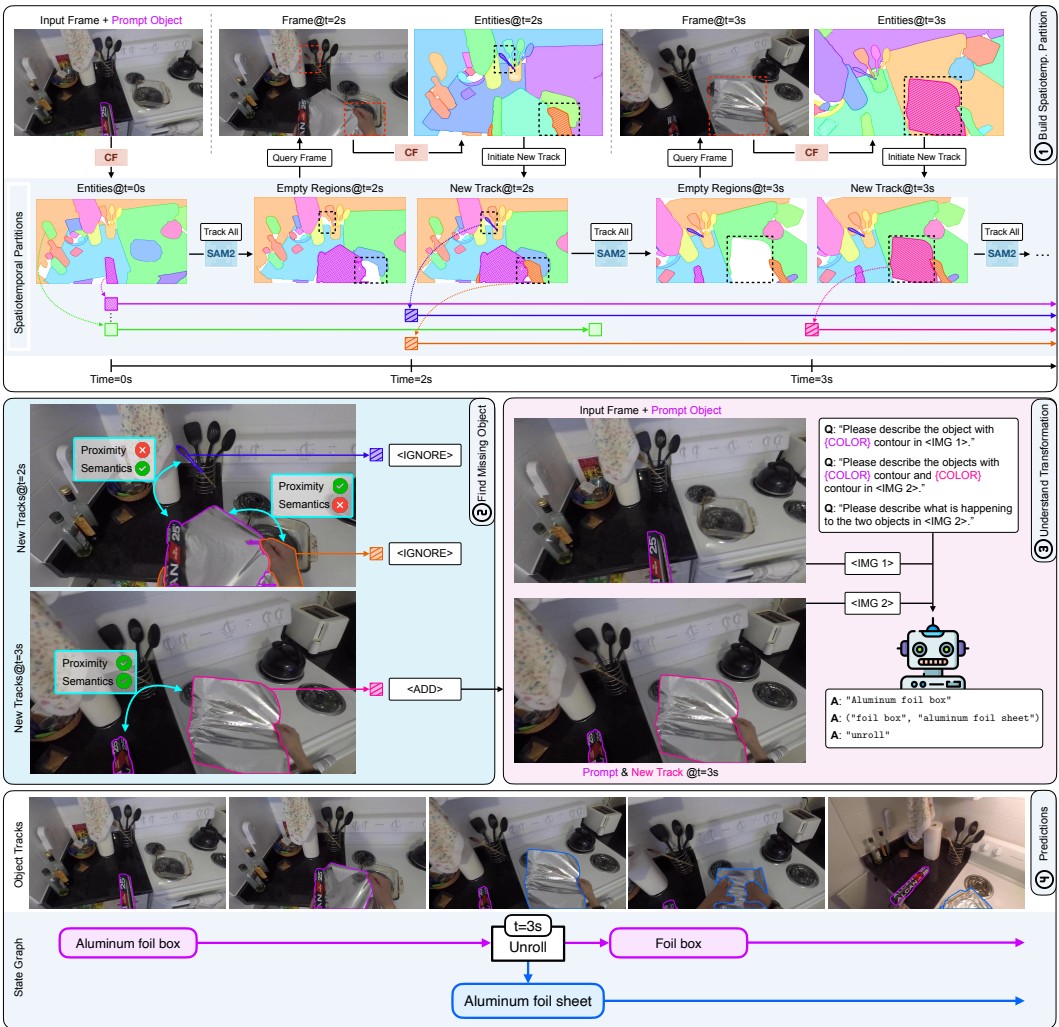

Figure 2: Overview of the proposed TubeletGraph. (1) Given a video and an initial prompt object mask, we first partition the initial frame via CropFormer (CF) [33] and track every region forward in time via SAM2 [35]. For each empty region at a later frame, we initiate a new track if an entity at that frame can match with it. In the end, we would obtain a spatiotemporal partition of the video. (2) For each later-emerged entity region, we reason about its proximity and semantic consistency with the prompt object and only recover regions that satisfy both. (3) For each recovered region, we prompt multi-modal LLMs to describe the transformation and resulting objects. (4) From this, TubeletGraph achieves consistent tracking of transformation objects while mapping every transformation and resulting regions in a state graph representation.

incrementally add new tubelets to $\mathcal{P}$ by initializing a track whenever track-less regions emerge. Concretely, we iterate through the frames and in every frame $t > 1$, we compute the entity segmentation $\mathcal{E}_t = \mathrm{CF}(I_t)$. For each entity $\hat{e}_t^j \in \mathcal{E}_t$, we initiate a new tubelet if less than $\tau_{\mathrm{coverage}}$ of its area is covered by an existing tubelet. This new tubelet $P'$ starting from entity $\hat{e}_t^j$ is then added to $\mathcal{P}$. This process ensures that the tubelets in $\mathcal{P}$ cover almost all of the pixels in the video.

The benefits of the spatiotemporal partition $\mathcal{P}$ are three-fold: (1) It forces every region in the video to be associated with some partition tubelet, maximizing the likelihood of object retrieval. (2) It reduces the complexity of the searching problem by reformulating a continuous problem of "where is the missing object in each frame" to a simpler discrete problem of "which partition tubelet is a real missing object." (3) It narrows down the set of candidate tubelets to only the ones added after the initial frame, since all initial tubelets in $\mathcal{P}_{\mathrm{init}}$ that are not the prompt can be immediately rejected.

## 3.3 Reasoning about New Candidate Entities

While $\mathcal{P}$ contains all tubelets that emerge after the initial frame, not every entity track discovered in a later frame is a real missing object. They can be new objects introduced in later frames, existing objects that are under-segmented in the initial entity segmentation, or missing products of the target object's state change that we wish to recover. Thus, we need a way to identify the latter from other irrelevant regions.

Here, we make two assumptions about object transformations in the real-world: (1) An object's location does not change drastically in a short period of time (e.g. an emerging butterfly is near its chrysalis), and (2) an object's identity and semantics cannot be significantly altered by transformations (a chrysalis can turn into a butterfly, but not a bird).

From this, we define two requirements that a candidate tubelet in $\mathcal{P}$ must satisfy to be considered as a missing object: spatial proximity and semantic consistency.

**Spatial Proximity.** By assuming temporally smooth object motions, the tubelets that were initiated near the prompt object track are more likely to be genuine missed objects. To estimate the spatial region where the transformed object might be located, we leverage the multiple candidate masks predicted by SAM2. These multiple masks $\{m_t^j\}_{j=1}^3$, originally intended to capture ambiguity in the user prompts, can also capture the ambiguity of prompt object segmentation during transformation. For a candidate track $C = \{c_s, c_{s+1}, ..., c_T\}$ that begins at frame $s$, and the prompt object track $P = \{p_1, p_2, ..., p_T\}$, we compute the following spatial proximity measure:

$$S_{\text{prox}}(C, P) = \max_{j \in \{1,2,3\}} |c_s \cap m_s^j| \, / \, |c_s| \tag{2}$$

where $\{m_s^j\}_{j=1}^3$ corresponds to the three candidate masks of prediction $p_s$. Intuitively, $S_{\text{prox}}$ captures the maximum overlap of the candidate track $C$ with any of $\{m_t^j\}_{j=1}^3$ at the frame where the candidate first appear. We consider a candidate proximal if $S_{\text{prox}}(C, P) > \tau_{\text{prox}}$.

**Semantic Consistency.** While the proximity prior eliminates candidate tracks that do not initiate nearby the prompt object, it is not sufficient by itself.

Consider the case of pulling out a sheet of foil (Figure 2, middle-left). The hands emerge in view holding the foil, but they should not be considered as the prompt object due to clear inconsistent semantics. To model this, we introduce a semantic consistency prior that assumes semantic alignment between a candidate entity and the prompt object. For a given mask $M$ and frame $I$, we compute the masked CLIP [34] feature $f(M, I) = \text{Pool}(\text{CLIP}(I), M)$ via mask-pooling [54].

For a candidate track $C = \{c_s, c_{s+1}, ..., c_T\}$ that begins at frame $s$, and the prompt object track $P = \{p_1, p_2, ..., p_T\}$, we compute the semantic similarity as:

$$S_{\text{sem}}(C, P) = \max_{i \in \{1,...,s-1\}, j \in \{s,...,T\}} f(p_i, I_i) \cdot f(c_j, I_j)^T \tag{3}$$

$S_{\text{sem}}$ captures the maximum pairwise similarity between the prompt track (prior to the candidate's emergence) and any mask in the candidate track. We consider a candidate semantically consistent if $S_{\text{sem}}(C, P) > \tau_{\text{sem}}$.

**Reasoning with Constraints.** By combining these two prior constraints, we only recover candidate tracks that are both semantically consistent and spatially proximal to the prompt object. As illustrated in Figure 2 (middle-left), this successfully removes false-positive candidates (e.g. cooking utensils and actor's hands) while retaining the genuine candidate (e.g. foil sheet).

Formally, we predict the set of valid continuation tracks as:

$$\mathcal{V} = \{C \in \mathcal{P} \mid C \text{ begins at } t > 0, \ S_{\text{prox}}(C, P) > \tau_{\text{prox}} \text{ and } S_{\text{sem}}(C, P) > \tau_{\text{sem}}\} \tag{4}$$

By combining $\mathcal{V}$ with the original prompt track $P$, we form the final tracking result $\mathcal{T}$ that successfully captures the complete prompt object through transformation.

## 3.4 Understanding Object Transformation

After recovering all candidate tracks that satisfy the two constraints, we leverage their emergence as indicators for when state transformation have occurred. For each valid continuation track $C \in \mathcal{V}$ that

Table 1: Object Tracking Performance on VOST [41] validation set. We compare multiple variants of TubeletGraph against base SAM2.1 and SAM2.1 (ft), which is finetuned on VOST train split. ST, S, and P indicate spatiotemporal partition, semantic consistent constraint, and spatial proximity constraint, respectively. $\mathcal{J}$ and $\mathcal{J}_{tr}$ measure tracking performance for the entire and last 25% of the video, while $\mathcal{P}$ and $\mathcal{R}$ measure per-pixel precision and recall.

| Method | ST | S | P | $\mathcal{J}$ | $\mathcal{J}^S$ | $\mathcal{J}^M$ | $\mathcal{J}^L$ | $\mathcal{P}$ | $\mathcal{R}$ | $\mathcal{J}_{tr}$ | $\mathcal{J}^S_{tr}$ | $\mathcal{J}^M_{tr}$ | $\mathcal{J}^L_{tr}$ | $\mathcal{P}_{tr}$ | $\mathcal{R}_{tr}$ |
|---|---|---|---|---|---|---|---|---|---|---|---|---|---|---|---|
| SAM2.1 (ft) | | | | 54.4 | 46.2 | 53.8 | 73.1 | 70.9 | 65.5 | 36.4 | 25.7 | 35.1 | 61.3 | 53.2 | 45.4 |
| SAM2.1 | | | | 48.4 | 40.7 | 50.0 | 63.0 | 71.3 | 54.5 | 32.4 | 21.2 | 34.3 | 54.2 | 58.5 | 34.7 |
| | ✓ | | | 25.7 | 22.4 | 27.2 | 30.6 | 18.6 | **71.5** | 13.9 | 9.8 | 14.1 | 22.8 | 10.8 | **59.4** |
| Ours | ✓ | ✓ | | 49.2 | 39.7 | 52.6 | 64.7 | 63.7 | 64.8 | 35.8 | 23.5 | 39.9 | 56.7 | 48.6 | 49.3 |
| (SAM2.1) | ✓ | | ✓ | 50.7 | **41.3** | **53.2** | 67.5 | 67.7 | 63.8 | 36.6 | **23.6** | **40.5** | 59.1 | 54.9 | 47.1 |
| | ✓ | ✓ | ✓ | **50.9** | **41.3** | 53.0 | **68.6** | **68.1** | 63.7 | **36.7** | **23.6** | 40.2 | **60.1** | **55.2** | 47.0 |

begins at frame $s$, we wish to know what transformation occurred and what are the resulting objects. As shown in Figure 2 (middle-right), we draw contours on the initial frame $I_1$ and frame $I_s$ and query multi-modal LLMs to provide a brief description for the transformation and object identity.

After parsing the natural language outputs, we construct the state graph as shown in Figure 2 (bottom). This state graph $\mathcal{S}$ provides a rich, structured representation of object transformations throughout the video, beyond consistently tracking the prompt object through transformation.

# 4 Experiments

**Datasets.** *VOST* [41] is curated from ego-centric videos in Ego4D [14] and EPIC-Kitchens [8] that contain object transformations from actor-object interactions. The validation set contains 70 videos with an average of 22.3 seconds captured at 60 fps, with 114 object masklets annotated at 5 fps. *VSCOS* [53] is constructed in a similar fashion from EPIC-Kitchens [8]. Its validation set contains 98 videos with an average of 7.5 seconds captured at 60 fps and object mask annotated at 1 fps. *M³-VOS* [5] models object phase changes and contains limited camera motion due to its source from online videos. The entire dataset serves as evaluation, containing 479 videos, 526 masklets, with an average of 14.3 seconds captured at 30 fps. Also, we evaluate on *DAVIS 2017* [32] to confirm tracking performance for objects that are not undergoing transformations.

*VOST-TAS (Track Any State):* We introduce a new benchmark for evaluating the proposed task by manually annotating transformations in the VOST [41] val set. Each object instance includes a list of transformations with temporal boundaries (start/end frames), action verb descriptions, and a list of resulting objects with segmentation masks and text descriptions on the end frame per transformation. In total, it contains 57 video instances, 108 transformations, and 293 annotated resulting objects.[2]

**Implementation Details.** For TubeletGraph, we adopt SAM2.1-L [35], CropFormer-Hornet-3X [33], FC-CLIP-COCO [54]. Hyperparameters for all three models are kept as default and not tuned further. In addition, we adopt GPT-4.1 [1] and keep sampling temperature at 0. To reason about new candidate entities, we select $\tau_{\text{prox}} = 0.3$ and $\tau_{\text{sem}} = 0.7$ after sweeping intervals of 0.1 on VOST train split that is similar sized as VOST val and applied to other datasets without any further modification. In addition, we arbitrarily ignore any entities smaller than $1/25^2$ of the video frame and set the coverage threshold for initiating new tracks $\tau_{\text{coverage}} = 0.25$ without further tuning.

## 4.1 Object Tracking

To measure object tracking performance, we follow VOST [41] and report Jaccard $\mathcal{J}$ and $\mathcal{J}_{tr}$ (only over last 25% frames), along with per-pixel precision $\mathcal{P}$ and recall $\mathcal{R}$. For a more fine-grain analysis, we divide each dataset into three equal subsets: small (S), medium (M), and large (L), based on the average object size throughout the video.

---

[2]Please refer to Appendix A.1 for details regarding VOST-TAS construction and visualization.

Table 2: Tracking Performance on VOST [41], VSCOS [53], M³-VOS [5], and DAVIS17 [32]. $\mathcal{J}$ and $\mathcal{J}_{tr}$ measure tracking performance for the entire and last 25% of the video, respectively. Best performance is bolded and second bests are underlined.

| Method | Detects + Describes Changes | VOST val | | VSCOS val | | M³-VOS val | | DAVIS17 val | |
|---|---|---|---|---|---|---|---|---|---|
| | | $\mathcal{J}$ | $\mathcal{J}_{tr}$ | $\mathcal{J}$ | $\mathcal{J}_{tr}$ | $\mathcal{J}$ | $\mathcal{J}_{tr}$ | $\mathcal{J}$ | $\mathcal{J}_{tr}$ |
| XMem [7] | ✗ | 36.1 | 24.7 | 69.9 | 64.6 | 69.7 | 60.1 | 82.9 | 81.0 |
| Cutie [6] | ✗ | 41.1 | 25.5 | 70.9 | 67.1 | 74.5 | 64.7 | 84.6 | 82.0 |
| ReVOS [5] | ✗ | 41.0 | 25.3 | - | - | **75.6** | **66.5** | 86.0 | 84.8 |
| SAM2 [35] | ✗ | 46.1 | 29.4 | 72.5 | 67.1 | 71.3 | 59.8 | 85.5 | 82.3 |
| SAM2Long [11] | ✗ | 46.4 | 29.1 | 73.0 | 68.6 | 70.2 | 58.7 | **87.1** | **85.5** |
| SAM2.1 [35] | ✗ | 48.4 | 32.4 | 72.0 | 66.9 | 71.3 | 59.3 | 85.7 | 83.0 |
| DAM4SAM [42] | ✗ | 48.8 | 33.6 | 71.3 | 66.0 | 72.2 | 61.3 | 86.2 | 84.2 |
| SAMURAI [48] | ✗ | 49.8 | 34.0 | 71.8 | 66.9 | 72.6 | 61.6 | 85.6 | 82.7 |
| Ours | ✓ | **50.9** | **36.7** | **75.9** | **72.2** | 74.1 | 64.1 | 85.6 | 82.6 |

**Analysis on VOST.** Table 1 presents the comparison between variants of TubeletGraph, the base SAM2 model, and SAM2 fintuned on the VOST training set. The top half of Table 1 first demonstrates an imbalanced error distribution for base SAM2: while precision $\mathcal{P}$ remains at over 70%, recall $\mathcal{R}$ languishes below 55%. This gap indicates that false negative errors are more than twice as frequent as false positives when tracking transforming objects, further confirming our observation that appearance-driven trackers struggles primarily with missed tracks than wrong tracks. As expected, finetuning SAM2 on the VOST yields substantial improvements across the board, with notable increase in recall $54.5$ to $65.6$ while maintaining precision. While finetuning shows clear benefits, it is limited by the extensive annotation cost for each specific transformation domain which reduces generalizability. In contrast, TubeletGraph is training-free, offering zero-shot capabilities and improved generalization.

The bottom half of Table 1 demonstrates the effectiveness of TubeletGraph. First, the proposed spatiotemporal partition is effective in providing candidate objects for retrieval. If every later-emerged object from the partition is incorporated into the prediction, the recall would greatly surpass that of the finetuned SAM2 model ($+6$ for $\mathcal{R}$ and $+14$ for $\mathcal{R}_{tr}$). However, as expected, this aggressive recovery comes at a cost of significant precision reduction ($-52.7$ for $\mathcal{P}$ and $-47.7$ for $\mathcal{P}_{tr}$).

By introducing the proposed semantic consistency and spatial proximity constraints, we can improve this precision-recall tradeoff. Notably, we are able to improve precision ($+49.5$ for $\mathcal{P}$ and $+44.4$ for $\mathcal{P}_{tr}$) while minimizing reduction in recall (-7.8 for $\mathcal{R}$ and -12.4 for $\mathcal{R}_{tr}$). While semantic prior brings marginal improvement when proximity prior is already considered, the consistent gain suggests its necessity (e.g., rejecting false positive entities that are close to the tracked object).

As a result, TubeletGraph is able to improve $\mathcal{J}$ by 2.5 points from the base SAM2 while *surpassing the finetuned SAM2 in* $\mathcal{J}_{tr}$. Finally, we obtain p-values of $0.014$ for $\mathcal{J}$ and $0.013$ for $\mathcal{J}_{tr}$ from a paired t-test between the base SAM2.1 and TubeletGraph, giving statistical significance to our improvement.

**Main Results.** Table 2 showcases a comprehensive comparison of tracking performance between our TubeletGraph and state-of-the-art baselines across four VOS benchmarks datasets[3]. Notably, TubeletGraph is the only method capable of not only tracking objects under transformations but also detecting and describing these state changes. Along with this additional capability, TubeletGraph achieves state-of-the-art performance on both VOST and VSCOS datasets, both focusing on transforming objects in ego-centric domains. When evaluated on M³-VOS, TubeletGraph outperforms all SAM-based variants and achieves results comparable to the best performing ReVOS. Finally, we measure all method performances on DAVIS17. Encouragingly, TubeletGraph performs comparably to all baselines, indicating that our approach of adding new tracks induces minimal false positives when tracking objects without transformations.

**Qualitative Results.** Figure 3 showcases our proposed system. Compared to prior works that miss object components due to transformations, TubeletGraph recovers the missing objects and leverage them as markers to describe the underlying transformation that caused the false negatives.

---

[3]Complete tracking results for VSCOS [53] and M³-VOS [5] are found in Appendix A.4

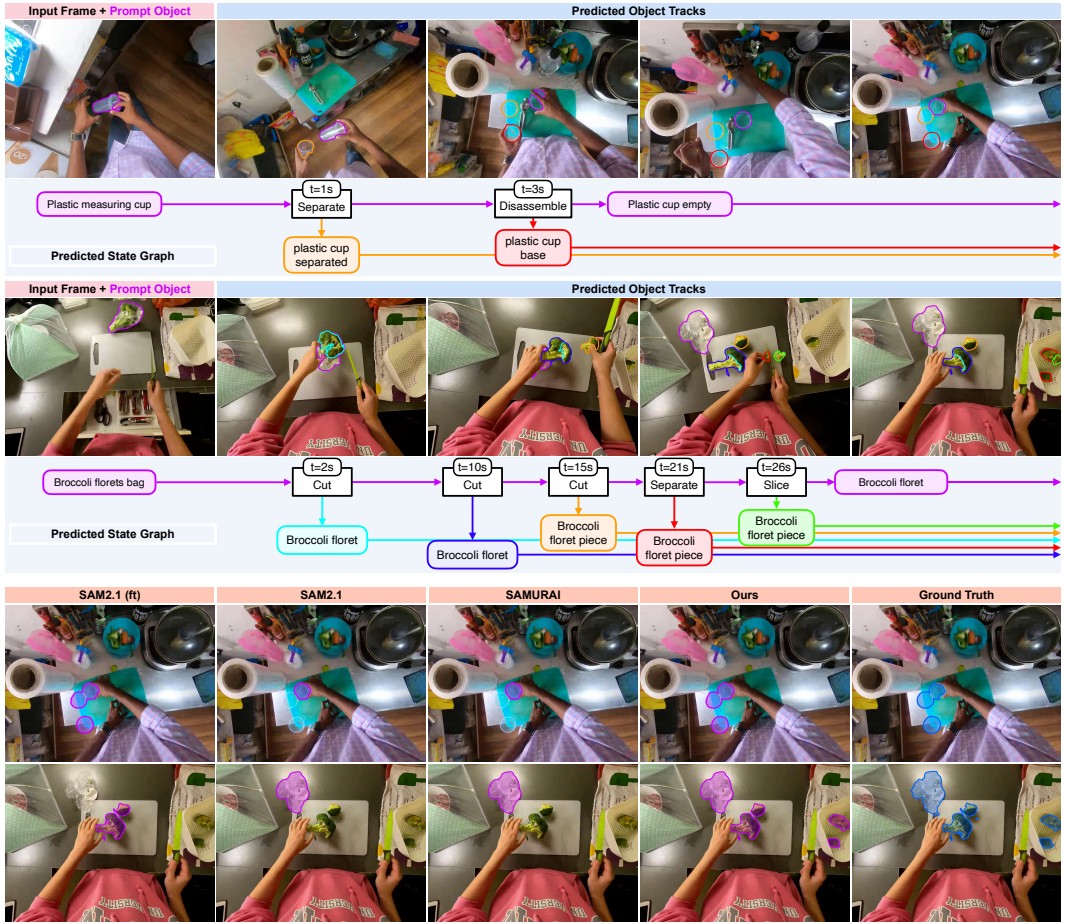

Figure 3: Qualitative Results on VOST val. We showcase TubeletGraph's tracking and state graph predictions on top, with comparisons against baselines at a particular ending frame at the bottom.

## 4.2 Transformation State Graph

To evaluate state graph quality, we report precision $\mathcal{T}_P$ and recall $\mathcal{T}_R$ for temporal localization within annotated transformation boundaries, and description accuracy for correctly localized action verbs ($\mathcal{A}_V$) and resulting objects ($\mathcal{A}_O$) with IoU $> 0.5$. Finally, we combine these into two metrics: spatiotemporal recall $\mathcal{H}_{ST}$ (correctly detects transformation within boundaries and finds all objects with IoU $> 0.5$) and overall recall $\mathcal{H}$ (additionally requiring correct action and object descriptions).[4]

**Temporal Localization.** We first report precision $\mathcal{T}_P$ and recall $\mathcal{T}_R$ for temporal localization. As shown in Table 3, TubeletGraph achieves $\mathcal{T}_P = 43.1$ and $\mathcal{T}_R = 20.4$ on VOST-TAS. While the precision is moderate, the relatively low recall indicates that many ground truth transformations are not detected within the annotated temporal boundaries. This stems from the passive detection of transformations, as they are only triggered when a false-negative object is recovered. For transformations that do not alter object appearances, accurate tracking would prevent transformation detection.

**Semantic Accuracy.** We then evaluate the semantic quality of predicted action verbs ($\mathcal{S}_V$) for transformations that are correctly localized temporally, and resulting object descriptions ($\mathcal{S}_O$) for objects that are matched with IoU $> 0.5$. As shown in Table 3, TubeletGraph achieves $\mathcal{S}_V = 81.8$ for action verbs and $\mathcal{S}_O = 72.3$ for resulting objects, demonstrating accurate description of the VLM-based reasoning module. Finally, since $\mathcal{S}_V$ and $\mathcal{S}_O$ are computed only on successful matches, they represent the description quality conditional on successful temporal/spatial localization.

---

[4]More details regarding metric computations are found in Appendix A.2.

Table 3: Object tracking ablation on VOST [41] and state graph results on VOST-TAS. $\mathcal{J}$ and $\mathcal{J}_{tr}$ measure tracking performance, $\mathcal{S}_V$ and $\mathcal{S}_O$ measure semantic accuracy of the state graph, $\mathcal{T}_P$ and $\mathcal{T}_R$ measure the precision and recall for temporal localization, while $\mathcal{H}_{ST}$ and $\mathcal{H}$ measures the combined transformation recall. $\mathcal{S}_V$ and $\mathcal{S}_O$ are shown in gray as the relative accuracies vary across methods.

| Entity | Tubelet Track | Semantic Sim. | VLM | Tracking | | Sem. Acc. | | Temp. Loc. | | Overall | |
|---|---|---|---|---|---|---|---|---|---|---|---|
| | | | | $\mathcal{J}$ | $\mathcal{J}_{tr}$ | $\mathcal{S}_V$ | $\mathcal{S}_O$ | $\mathcal{T}_P$ | $\mathcal{T}_R$ | $\mathcal{H}_{ST}$ | $\mathcal{H}$ |
| SAM [19] | ✓ | ✓ | ✓ | 49.2 | 34.7 | 81.0 | 72.9 | 36.8 | 19.4 | 10.2 | 0.9 |
| ✓ | Cutie [6] | ✓ | ✓ | 47.6 | 33.9 | 91.7 | 80.6 | 32.4 | 11.1 | 6.5 | 2.8 |
| ✓ | ✓ | DINOv2 [30] | ✓ | 50.9 | 36.6 | 76.2 | 77.1 | 42.0 | 19.4 | 12.0 | 5.6 |
| ✓ | ✓ | ✓ | Qwen [2] | 50.9 | 36.7 | 31.8 | 44.6 | 43.1 | 20.4 | 12.0 | 1.9 |
| CF [33] | SAM2.1 [35] | CLIP [54] | GPT4.1 [1] | 50.9 | 36.7 | 81.8 | 72.3 | 43.1 | 20.4 | 12.0 | 6.5 |

**Overall Performance.** Finally, we compute spatiotemporal recall $\mathcal{H}_{ST}$ (correct temporal localization with every resulting object matched with IoU $> 0.5$) and overall recall $\mathcal{H}$ (additionally requiring all correct semantic descriptions). As shown in Table 3, TubeletGraph achieves $\mathcal{H}_{ST} = 12.0$ and $\mathcal{H} = 6.5$. *This reflects the significant difficulty of transformation prediction in unconstrained ego-centric videos.* As the first approach to jointly tackle object tracking and state graph prediction, these results establish a baseline for Track Any State and highlight clear directions for future work.

### 4.3 System Analysis and Discussion

**Robustness.** We first systematically ablate each component by while keeping other modules fixed (Table 3). When replacing CropFormer [33] with SAM automasks [19] for entity detection, tracking performance reduces by 1.7 in $\mathcal{J}$, which is mainly attributed to SAM being less reliable for small objects (Table 4). Replacing SAM2.1 [35] with Cutie [6] for tubelet propagation results in a more significant degradation ($-3.3$ in $\mathcal{J}$ and $-9.3$ in $\mathcal{T}_R$), indicating the importance of accurate tubelet tracking. For semantic filtering, swapping CLIP [54] with DINOv2 [30] yields comparable tracking performance. Finally, replacing GPT-4.1 [1] with Qwen-2.5VL [2] dramatically impacts semantic accuracy, demonstrating high-quality VLM reasoning is critical for accurate semantic descriptions.

Additionally, we find TubeletGraph to be highly robust to the filtering thresholds $\tau_{\mathrm{prox}}$ and $\tau_{\mathrm{sem}}$. On $M^3$-VOS and VSCOS, we obtain a robust range of $(72.6, 74.2)$ and $(75.1, 75.9)$ for $\mathcal{J}$, respectively, after sweeping $\tau_{\mathrm{prox}}$ between $0.1$ and $0.5$ and $\tau_{\mathrm{sem}}$ between $0.5$ and $0.9$ in intervals of $0.1$ (Table 6).

**Computational Efficiency.** The main efficiency bottleneck of TubeletGraph is constructing a spatiotemporal partition by tracking every spatial region, which costs on average 7 seconds per frame on VOST [41] with one NVIDIA RTX A6000 GPU. Although limiting real-time applications, TubeletGraph's unique capabilities to track objects while detecting and describing transformations can be very useful; e.g., producing training annotations on recorded demonstrations for robots, analyzing compliance videos on the factory floor, understanding animal developments from camera traps. In these applications, understanding and tracking object transformations is critical, and real-time is not needed. Finally, the spatiotemporal partition can be adapted to multi-object tracking with little-to-no additional cost, amortizing the computational time when tracking multiple objects simultaneously.

## 5 Conclusion

In this work, we introduce the problem of Track Any State, tracking objects through transformations while detecting and describing the transformation. We proposed TubeletGraph, a zero-shot system that recovers missing objects after transformation and leverage them as "landmarks" to reason and describe them. Our approach achieves state-of-the-art tracking performance under transformation while demonstrating promising capabilities in spatiotemporal grounding of object transformations.

**Limitations and Broader impacts.** Beyond high computational cost, the modular design of Tubelet-Graph may pose potential challenges for systematic error attribution and diagnosis. Finally, our work does not introduce any foreseeable societal impacts, but will generally promote more robust and informative tracking systems for robotics and general vision systems. [5]

---

[5] Please refer to Appendix A.5 for additional discussions on error analysis and broader impacts.

## 6 Acknowledgement

This research is based upon work supported in part by the National Science Foundation (IIS-2144117, IIS-2107161 and IIS-2505098). Yihong Sun is supported by an NSF graduate research fellowship.

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

# A  Appendix

## A.1  VOST-TAS Dataset

**Dataset Overview.**  We introduce VOST-TAS (TrackAnyState), an extended version of the VOST validation set [41] with explicit transformation annotations. Given a video sequence $V = \{I_t\}_{t=0}^{T}$ where $I_t$ denotes the frame at time $t$, we annotate each temporal segments corresponding to object state transformations. In total, VOST-TAS contains 57 video instances, 108 transformations, and 293 annotated resulting objects. Qualitative examples are provided in Figure 4 and the full dataset is available at https://github.com/YihongSun/TubeletGraph.

**Dataset Details.**  For each video instance, we manually label a corresponding annotation $A = (t_{\text{start}}, t_{\text{end}}, \Gamma)$. Here, $t_{\text{start}} = 0$ denotes the initial annotation frame; $t_{\text{end}} \in [0, T]$ denotes the terminal annotation frame; and $\Gamma = \{\tau_i\}_{i=1}^{N}$ containing the set of $N$ transformations.

In addition, each transformation $\tau_i$ is formally represented as a tuple as $\tau_i = (t_i^{\text{s}}, t_i^{\text{e}}, v_i, \mathcal{O}_i)$. Here, $t_i^{\text{s}}, t_i^{\text{e}} \in [t_{\text{start}}, t_{\text{end}}]$ define the temporal start/end boundaries of the transformation; $v_i$ contains the free-text descriptions of the transformation; and $\mathcal{O}_i = \{(M_{i,j}, d_{i,j})\}_{j=1}^{K_i}$ is the set of $K_i$ resulting objects, where $M_{i,j}$ denotes the segmentation mask and $d_{i,j}$ the textual description, both annotated at frame $t_i^{\text{e}}$.

**Annotation Protocol.**  We employ the following criteria to ensure consistency and quality:

(1) *Physical Separability*: Resulting objects are considered distinct if they are physically separable, even if visually contiguous and semantically identical.

(2) *Diversity Constraint*: To allow diversity in the transformations, annotation process terminates when an action-object pair $(v, o)$ occurs more than three times in a row. Similarly, duplicated objects in the same video with identical descriptions and associated actions are excluded. The early-stopped annotation would lead to terminal annotation frame $t_{\text{end}} < T$.

(3) *Quality Filtering*: Transformations are excluded if the target object is not clearly visible during transformation or if the state change is ambiguous.

## A.2  Track Any State Evaluation Metrics

We evaluate state graph quality across two dimensions: temporal localization and semantic accuracy on the VOST-TAS Dataset.

**Temporal Localization Metrics.**  To assess the temporal localization of predicted transformations, we measure precision $\mathcal{T}_P$ and recall $\mathcal{T}_R$ using bipartite matching between predicted timestamps and ground truth temporal ranges.

Given a video instance with ground truth annotation $(t_{\text{start}}, t_{\text{end}}, \Gamma)$ and transformation intervals $\Gamma = \{(t_i^{\text{s}}, t_i^{\text{e}}, v_i, \mathcal{O}_i)\}_{i=1}^{N_g}$ containing $N_g$ transformations, we obtain all predicted transformation timestamps between $t_{\text{start}}$ and $t_{\text{end}}$, denoted as $\mathcal{P} = \{t_j^{\text{pred}}\}_{j=1}^{N_p}$. From this, we construct a cost matrix $C \in \{0, 1\}^{N_g \times N_p}$ where

$$C_{ij} = \begin{cases} 0 & \text{if } t_j^{\text{pred}} \in [t_i^{\text{s}}, t_i^{\text{e}}] \\ 1 & \text{otherwise} \end{cases} \tag{5}$$

We then apply the Hungarian algorithm [20] to find the optimal assignment minimizing total cost. Predictions matched with cost 0 are counted as true positives (TP), while unmatched predictions contribute to false positives (FP) and unmatched ground truths to false negatives (FN). Precision $\mathcal{T}_P$ and recall $\mathcal{T}_R$ are then computed as:

$$\mathcal{T}_P = \frac{\text{TP}}{\text{TP} + \text{FP}}, \quad \mathcal{T}_R = \frac{\text{TP}}{\text{TP} + \text{FN}} \tag{6}$$

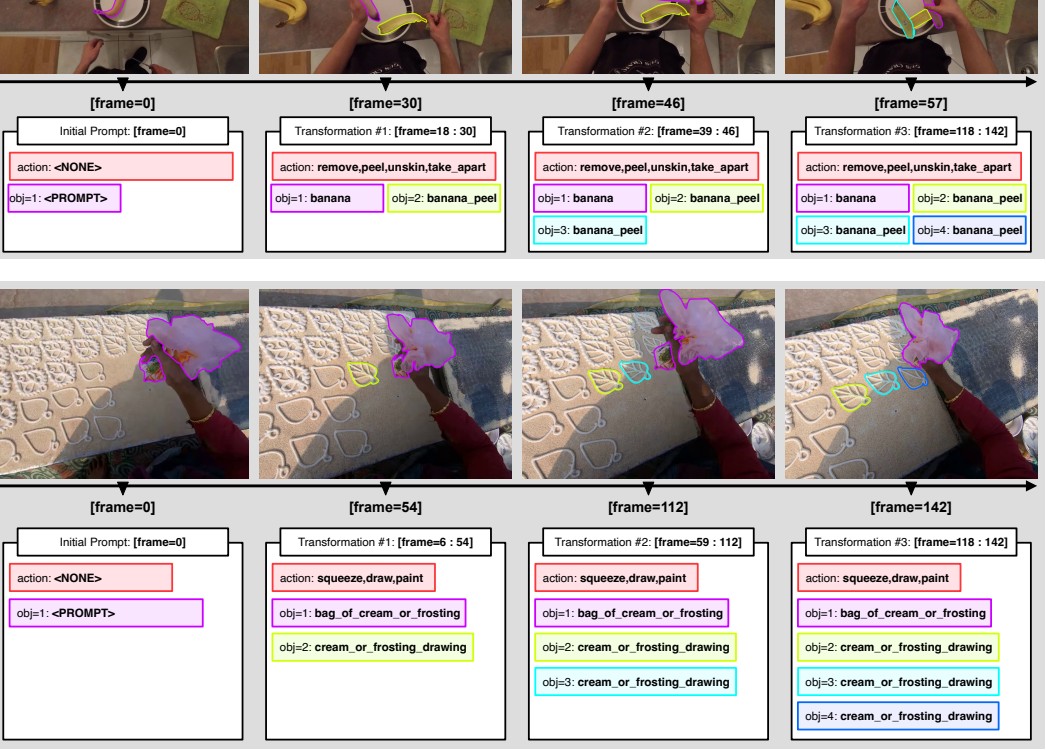

Figure 4: Examples of VOST-TAS.

**Semantic Accuracy Metrics.** Beyond temporal localization, we also evaluate the semantic quality of predicted action verbs and resulting objects.

*Action Verb Accuracy ($\mathcal{A}_V$):* For any predicted transformation that is correctly matched to a ground truth temporal boundary (i.e. $t_j^{\text{pred}} \in [t_i^{\text{s}}, t_i^{\text{e}}]$), we assess whether the predicted action descriptions semantically match ground truth description using GPT-4.1 [1] with temperature $= 0$ for deterministic evaluation. The model receives the system prompt:

> *"You are a highly intelligent assistant that can analyze actions in text."*

followed by the evaluation prompt:

> *"Given a particular action description of '[GT_ACTION]', is '[PRED_ACTION]' similar to the verbs in this action? Please rate from -1 to 1, where -1 means completely unrelated, 0 means ambiguous, and 1 means '[PRED_ACTION]' captures the meaning of '[GT_ACTION]' or is directly in it. Brief/general descriptions should still be considered as +1. Please answer with a single integer."*

A prediction is considered correct if the model returns a score of 1.

*Resulting Object Accuracy ($\mathcal{A}_O$):* For any predicted transformation that is correctly matched to a ground truth temporal boundary (i.e. $t_j^{\text{pred}} \in [t_i^{\text{s}}, t_i^{\text{e}}]$), we perform Hungarian matching [20] on the IoU matrix between predicted masks and ground truth masks and collect all matched pairs with IoU $> 0.5$ for semantic evaluation. For each spatially matched objects, we evaluate description similarity using GPT-4.1 with the system prompt:

> *"You are a highly intelligent assistant that can analyze actions and resulting objects in text."*

and the evaluation prompt:

*"Given the object description '[GT_OBJECT]', is '[PRED_OBJECT]' similar to it? Please rate from -1 to 1, where -1 means completely unrelated, 0 means ambiguous, and 1 means '[PRED_OBJECT]' is similar. Over- or under-specified descriptions should still be considered as +1. Please answer with a single integer."*

A prediction is considered correct if the model returns a score of 1.

**Combined Metrics.** Finally, we define two holistic metrics combining temporal and semantic evaluation:

- **Spatiotemporal Recall** ($\mathcal{H}_{ST}$)**:** For each ground truth transformation $\tau_i = (t_i^s, t_i^e, v_i, \mathcal{O}_i)$ in a video instance, a prediction is considered a spatiotemporal match if: (1) the predicted timestamp $t_j^{\text{pred}}$ is correctly matched within $[t_i^s, t_i^e]$, and (2) all $K_i$ ground truth resulting objects in $\mathcal{O}_i$ are matched with predicted masks with IoU $> 0.5$ at frame $t_i^e$. From this, the spatiotemporal recall $\mathcal{H}_{ST}$ is computed as:

$$\mathcal{H}_{ST} = \frac{\text{\# of spatiotemporally matched transformations}}{\text{\# of ground truth transformations}} \tag{7}$$

- **Overall Recall** ($\mathcal{H}$)**:** Building upon $\mathcal{H}_{ST}$, a transformation is considered fully correct if it satisfies all spatiotemporal matching criteria and additionally: (1) the predicted action description achieves $\mathcal{A}_V = 1$ (semantic match with ground truth action), and (2) all spatially matched resulting objects achieve $\mathcal{A}_O = 1$ (semantic match with ground truth object descriptions). Overall recall is computed as:

$$\mathcal{H} = \frac{\text{\# of fully correct transformations}}{\text{\# of ground truth transformations}} \tag{8}$$

### A.3 Additional Implementation Details

Shown in Section 3.2, we compute the complete spatial partition $\mathcal{E}_1$ for the initial frame $I_1$ as follows:

$$\mathcal{E}_1 = \text{CF}(I_1) \cup \{\mathcal{M}_1\} \tag{9}$$

Naturally, combining the object prompt $\mathcal{M}_1$ with the set of masks predicted by CropFormer(CF) [33] is not trivial. $\mathcal{M}_1$ can overlap with a subset of masks in $\text{CF}(I_1)$ at various degrees.

To resolve possible overlaps, we first denote the fraction of mask $a$ covered by mask $b$ as $\text{cover}(a, b)$. Then, we introduce another coverage threshold $\tau_{\text{remove}}$ (where $\tau_{\text{remove}} > \tau_{\text{coverage}}$) to remove any entity $e_1^i \in \text{CF}(I_1)$ with $\text{cover}(e_1^i, \mathcal{M}_1) \geq \tau_{\text{remove}}$.

Concretely, from the predicted entities $\text{CF}(I_1) = \{e_1^1, e_1^2, \ldots, e_1^n\}$, prompt mask $\mathcal{M}_1$, and coverage thresholds $\tau_{\text{coverage}}$ and $\tau_{\text{remove}}$, we construct two subsets $\mathcal{E}_1^{\text{keep}}$ and $\mathcal{E}_1^{\text{modify}}$ from $\text{CF}(I_1)$ as follows:

1. **Keep as-is**: For every predicted entity $e_1^i \in \text{CF}(I_1)$ with $\text{cover}(e_1^i, \mathcal{M}_1) < \tau_{\text{coverage}}$, we include it in $\mathcal{E}_1^{\text{keep}}$ without any modification.

$$\mathcal{E}_1^{\text{keep}} = \{e_1^i : e_1^i \in \text{CF}(I_1), \text{cover}(e_1^i, \mathcal{M}_1) < \tau_{\text{coverage}}\}$$

Table 4: Object Tracking Ablation Results on VOST [41] validation set. $\mathcal{J}$ and $\mathcal{J}_{tr}$ measure tracking performance for the entire and last 25% of the video, while $\mathcal{P}$ and $\mathcal{R}$ measure per-pixel precision and recall.

| Entity | Tubelet Track | Semantic | $\mathcal{J}$ | $\mathcal{J}^S$ | $\mathcal{J}^M$ | $\mathcal{J}^L$ | $\mathcal{P}$ | $\mathcal{R}$ | $\mathcal{J}_{tr}$ | $\mathcal{J}_{tr}^S$ | $\mathcal{J}_{tr}^M$ | $\mathcal{J}_{tr}^L$ | $\mathcal{P}_{tr}$ | $\mathcal{R}_{tr}$ |
|---|---|---|---|---|---|---|---|---|---|---|---|---|---|---|
| SAM Automask | ✓ | ✓ | 49.2 | 40.9 | 47.0 | **70.2** | 67.9 | 61.0 | 34.7 | 20.9 | 35.9 | **63.1** | 54.0 | 43.5 |
| ✓ | Cutie | ✓ | 47.6 | 40.8 | 43.7 | 67.9 | 66.9 | 59.0 | 33.9 | 22.3 | 31.8 | 62.1 | 53.6 | 41.0 |
| ✓ | ✓ | DINOv2 | 50.9 | **41.3** | 52.4 | 69.2 | **68.3** | 63.2 | 36.6 | **23.6** | 39.4 | 60.8 | **55.4** | 46.5 |
| CF | SAM2.1 | CLIP | **50.9** | **41.3** | **53.0** | 68.6 | 68.1 | **63.7** | **36.7** | **23.6** | **40.2** | 60.1 | 55.2 | **47.0** |

Table 5: Object Tracking Performance on VSCOS [53] and $M^3$-VOS [5] validation set. We compare multiple variants of our model against base SAM2.1 and SAM2.1 (ft), which is finetuned on VOST train split. ST, S, and P indicate spatiotemporal partition, semantic consistent constraint, and spatial proximity constraint, respectively. $\mathcal{J}$ and $\mathcal{J}_{tr}$ measure tracking performance for the entire and last 25% of the video, while $\mathcal{P}$ and $\mathcal{R}$ measure per-pixel precision and recall.

| Method | ST | S | P | $\mathcal{J}$ | $\mathcal{J}^S$ | $\mathcal{J}^M$ | $\mathcal{J}^L$ | $\mathcal{P}$ | $\mathcal{R}$ | $\mathcal{J}_{tr}$ | $\mathcal{J}_{tr}^S$ | $\mathcal{J}_{tr}^M$ | $\mathcal{J}_{tr}^L$ | $\mathcal{P}_{tr}$ | $\mathcal{R}_{tr}$ |
|---|---|---|---|---|---|---|---|---|---|---|---|---|---|---|---|
| | | | | | | VSCOS [53] | | | | | | | | | |
| SAM2.1 (ft) | | | | 79.8 | 77.0 | 78.9 | 83.4 | 91.8 | 85.5 | 78.3 | 73.0 | 78.2 | 83.8 | 90.7 | 85.4 |
| SAM2.1 | | | | 72.0 | 61.7 | 76.7 | 77.9 | 90.0 | 76.6 | 66.9 | 53.5 | 72.1 | 75.4 | 88.8 | 71.6 |
| | ✓ | | | 42.5 | 32.9 | 47.6 | 47.2 | 44.9 | **86.5** | 38.2 | 29.0 | 42.5 | 43.4 | 38.6 | **86.4** |
| Ours | ✓ | ✓ | | 75.5 | **68.1** | 78.0 | 80.4 | 87.1 | 84.0 | 71.5 | **61.5** | 75.5 | 77.6 | 85.2 | 82.8 |
| (SAM2.1) | ✓ | | ✓ | **75.9** | 67.8 | 79.0 | 80.9 | 89.1 | 83.0 | 72.1 | 60.7 | 77.4 | **78.4** | 87.0 | 81.8 |
| | ✓ | ✓ | ✓ | **75.9** | 67.8 | **79.1** | **81.0** | **89.3** | 82.9 | **72.2** | 60.7 | **77.6** | **78.4** | **87.4** | 81.7 |
| | | | | | | $M^3$-VOS [5] | | | | | | | | | |
| SAM2.1 (ft) | | | | 74.4 | 67.6 | 79.3 | 78.3 | 88.5 | 79.6 | 64.8 | 57.1 | 70.4 | 69.3 | 82.5 | 71.5 |
| SAM2.1 | | | | 71.3 | 66.8 | 74.7 | 73.9 | 89.7 | 75.1 | 59.3 | 54.4 | 62.0 | 62.8 | 84.8 | 63.5 |
| | ✓ | | | 60.9 | 48.8 | 63.1 | 74.5 | 66.6 | **84.0** | 51.4 | 38.1 | 53.8 | 66.4 | 58.6 | **79.3** |
| Ours | ✓ | ✓ | | 72.5 | 65.5 | 76.2 | 77.9 | 85.0 | 81.0 | 62.5 | 54.3 | 66.1 | 69.4 | 77.8 | 73.3 |
| (SAM2.1) | ✓ | | ✓ | 74.0 | **67.4** | 78.5 | 78.1 | 88.2 | 79.9 | **64.2** | 55.8 | 68.4 | **70.9** | 82.2 | 71.7 |
| | ✓ | ✓ | ✓ | **74.1** | **67.4** | **78.7** | **78.2** | **88.4** | 79.8 | 64.1 | **55.9** | **68.5** | 70.3 | **82.4** | 71.5 |

Table 6: Parameter sweep for semantic similarity threshold ($\tau_{\text{sem}}$) and proximity threshold ($\tau_{\text{prox}}$) for $\mathcal{J}$. Best performance is bolded. The parameter setting tuned on the VOST train ($\tau_{\text{sem}} = 0.7$, $\tau_{\text{prox}} = 0.3$) is found underlined in center grid.

| | $M^3$-VOS [5] | | | | | VSCOS [53] | | | | |
|---|---|---|---|---|---|---|---|---|---|---|
| $\tau_{\text{prox}} =$ | 0.1 | 0.2 | 0.3 | 0.4 | 0.5 | 0.1 | 0.2 | 0.3 | 0.4 | 0.5 |
| $\tau_{\text{sem}} = 0.5$ | 74.1 | 74.1 | 74.0 | 74.0 | 74.0 | 75.5 | 75.8 | **75.9** | 75.8 | 75.8 |
| $\tau_{\text{sem}} = 0.6$ | 74.1 | 74.1 | 74.0 | 74.1 | 74.0 | 75.5 | 75.8 | **75.9** | 75.8 | 75.8 |
| $\tau_{\text{sem}} = 0.7$ | **74.2** | **74.2** | 74.1 | 74.1 | 74.0 | 75.7 | 75.8 | **75.9** | 75.8 | 75.8 |
| $\tau_{\text{sem}} = 0.8$ | 73.7 | 73.6 | 73.6 | 73.6 | 73.6 | 75.5 | 75.6 | 75.7 | 75.7 | 75.7 |
| $\tau_{\text{sem}} = 0.9$ | 72.7 | 72.6 | 72.6 | 72.6 | 72.6 | 75.1 | 75.1 | 75.1 | 75.1 | 75.1 |

2. **Modify and remove overlap**: For every predicted entity $e_1^i \in \text{CF}(I_1)$ with $\tau_{\text{coverage}} \leq \text{cover}(e_1^i, \mathcal{M}_1) < \tau_{\text{remove}}$, we only include the non-overlapping component of $e_1^i$ in $\mathcal{E}_1^{\text{modify}}$.

$$\mathcal{E}_1^{\text{modify}} = \{e_1^i \setminus (e_1^i \cap \mathcal{M}_1) : e_1^i \in \text{CF}(I_1), \tau_{\text{coverage}} \leq \text{cover}(e_1^i, \mathcal{M}_1) < \tau_{\text{remove}}\}$$

Thus, we obtain the final $\mathcal{E}_1 = \mathcal{E}_1^{\text{keep}} \cup \mathcal{E}_1^{\text{modify}} \cup \{\mathcal{M}_1\}$.

## A.4 Additional Evaluations

Please refer to Table 5 for comparison results on $M^3$-VOS and VSCOS. We observe largely consistent trends as found in Table 1, which underlines the generalizability of TubeletGraph. In addition, Table 4 provides the full tracking results that are omitted in Table 3 due to space constraints. Since the use of different VLM models does not impact tracking performance, the ablation with Qwen [2] is omitted. Finally, Table 6 shows the full grid search over both spatial proximity and semantic consistency thresholds on $M^3$-VOS [5] and VSCOS [53]. The parameters tuned on the VOST train ($\tau_{\text{sem}}$=0.7, $\tau_{\text{prox}}$=0.3), found in center grid) perform competitively across all datasets. This verifies that the hyperparameter selection generalizes well across datasets without requiring dataset-specific tuning.

This robustness further underlines the stability of our filtering mechanism without a need for precise threshold calibration.

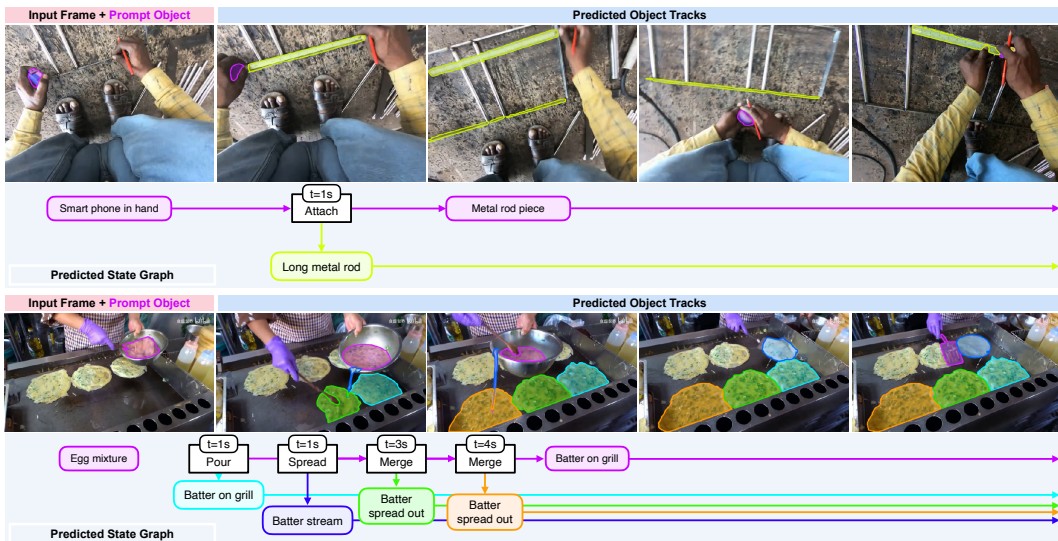

Figure 5: Failure examples of TubeletGraph.

## A.5 Additional Discussions.

**Failure Examples and Error Analysis** We first show failure examples of TubeletGraph in Figure 5. In the top example, the tape measure case is incorrectly identified as a "smartphone," and the extended measuring tape is described as a "metal rod." Consequently, the transformation is described as "attach" rather than the correct "extend" or "pull out." This failure stems from the incomplete object views in the selected frames passed to the VLM, where hand occlusions prevent accurate object recognition and lead to cascading error for the action description. In the bottom example, a false positive omelet is incorporated into the tracking of an resulting object in the state-graph. Specifically, the object track for the correctly identified "batter stream" incorrectly laches on to the irrelevant omelet later into the video. By assuming a high precision underlying tracker, TubeletGraph fails to remove this false positive error made by SAM2 [35]. Video failure examples can be found in https://tubelet-graph.github.io.

More generally, we found errors typically manifest as (1) false positive predictions by the base tracker and (2) minor reduction in tracking recall when applying semantic and proximal constraints as shown in Table 1. False positive errors (1) can cause erroneous measures of semantic and proximal similarities, while reduction in tracking recall can reduce recall for temporal localization of object transformations.

**Broader Impacts.** TubeletGraph's ability to track objects through transformations and describe state changes has broad applications across multiple domains. In robotics, it enables learning from demonstration by automatically annotating object state changes in recorded manipulation tasks, reducing the manual annotation burden for training data collection. In scientific research, it facilitates the study of developmental processes (e.g., tracking metamorphosis in insects, growth of cell cultures) from video recordings where manual annotation would be prohibitively expensive.

As with any technology that analyzes visual data, risks arise when applied to human behavior or in surveillance contexts. Understanding object transformations could potentially be misused to monitor individuals' activities in private settings without consent, or to enforce overly intrusive workplace surveillance that violates workers' privacy and dignity. In ego-centric applications particularly, the system processes first-person video that may inadvertently capture sensitive personal information or the activities of bystanders who have not consented to being recorded.

**Combining TubeletGraph with SAM2.1 (ft).** We find that integrating SAM2.1(ft) with Tubelet-Graph ($\mathcal{J} = 54.1$) shows modest improvements over the base TubeletGraph ($\mathcal{J} = 50.9$). However, the improvement is smaller than expected, given the strong standalone performance of SAM2.1(ft). We reason that is because TubeletGraph specifically addresses false-negative predictions by incorporating new candidate tracks lost due to object transformation. Since SAM2.1(ft) is fine-tuned on VOST to minimize these false negatives, the complementary benefits are naturally reduced.

**Effects of Occlusions on TubeletGraph.** During occlusion events, TubeletGraph would generally add additional tubelets for the target object that re-emerges after being temporarily lost. If this additional tubelet matches the semantic and proximity constraints, it will be incorporated in the tracked object. Since the proximity constraint relies on the candidate masks predicted by SAM2, the base tracker's internal candidate masks would set an upperbound for the object recovery after occlusion. In terms of the transformation detection, since the VLM only observes the frames where the objects are visible, the transformation would not be described as an occlusion event.

