# OpenReview forum: "Tracking and Understanding Object Transformations"
_NeurIPS.cc/2025/Conference — NeurIPS 2025 poster_

### Official Review · Reviewer_6G8j · 2025-07-03

**Clarity:** 3
**Significance:** 2
**Originality:** 3
**Rating:** 5
**Confidence:** 4

**Summary:**

The paper proposes a zero-shot framework called TubeletGraph that can track objects through transformations, identifying potential tracks after state changes and also reasoning about the state change through a VLM. Introducing Track Any State as a new problem definition, TubeletGraph creates a full state graph for the tracked object through extreme transformations where conventional trackers can commonly fail. The method performs favorable to other methods it compares against.

**Questions:**

- From the ablations it looks like once you have the spatiotemporal partition + proximity constraint, the semantic consistency brings little improvement. Do the authors think this potentially is a factor due to the dataset, and that semantic constraint can show more significant improvement in different cases?
- For reasoning, did the author’s try any open source models? I think it can be interesting to see if that changes the performance of reasoning and would make this framework more adoptable.
- Can the authors elaborate further on the fine-tuning on SAM2 that you made during ablations? The paper mentions clear high annotation cost for object transformations, do you mean the existing annotations in the dataset, or did you annotate anything further?

**Ethical Concerns:**

["NO or VERY MINOR ethics concerns only"]

**Final Justification:**

After the rebuttals to all reviews and the detailed answers the authors provided, I decided to increase my rating. I think the added discussions on semantic evaluation, the robustness to thresholds/different models have provided more strength to the proposed framework. Once concern remaining is that some revisions promised by the authors can be involved and might not be ready before camera-ready (e.g. extending evaluations to the full VOST val set). However, due to the detailed answers during the rebuttal, I am hoping these will be prioritized for a stronger final paper.

**Limitations:**

yes

**Paper Formatting Concerns:**

I don't see any major issues.

**Quality:**

3

**Strengths And Weaknesses:**

Strengths:
- The paper is well-written with an interesting problem covering very complex cases in object tracking.
- To my knowledge, the paper proposes a new paradigm in Track Any State.
- The experimental results show that for most cases, the proposed method that is zero-shot can show improvements over compared baselines/models.
Weaknesses:
- Creating the tubelets seems to be a very expensive process and not quite ready for practical use. (Seems like this is also discussed in the limitations) There's also several models involved, including a non-open source model (question on that below).
- It would be interesting to see how this zero-shot method performs under other common confounding factors in object tracking such as occlusions. In these kinds of conditions, do we just generate additional tubelets if object is temporarily lost? Do we expect VLM to reason the transformation is the occlusion? Since occlusions are a fundamental issue with tracking I think it would make the paper stronger to discuss this.
- Authors show ReVOS outperforms their method on $M^{3}$-VOS. It would be great to give more insights of why this is the only dataset where they don’t show superior performance in the Main Results section. Is it related to a property of the evaluation set that makes ReVOS especially strong on this set? Same with DAVIS17 results.
- The temporal localization is evaluated only on 8 manually annotated videos, this seems insufficient to me. In addition, the experiment is a little confusing, it might be worth it to explain further how the varying T values were chosen for each line in the graph.

---

> ### Author Rebuttal · Authors · 2025-07-31
>
> We thank Reviewer 6G8j for their positive evaluation and for recognizing the motivation and novelty of our work. Below, we address the reviewer’s specific questions and concerns.
>
> ---
> ### **W1. High computational cost**
>
> We acknowledge the concern regarding computational cost and agree that the current speed limits real-time applications. However, we believe TubeletGraph’s unique capabilities provide significant value despite this limitation.
>
> **Value Beyond Real-Time Performance**: TubeletGraph enables detecting and describing object state transformations beyond object tracking in videos. No existing methods perform both temporal localization and semantic description of the transformation in addition to state-of-the-art performance on object tracking under transformation. The combination of tracking and discovering, localizing and describing transformations from videos can be very useful for many applications: for example, producing training annotations on recorded demonstrations for robots, analyzing compliance and processes on the factory floor from videos, understanding animal behavior and development from camera traps and so on. In these applications, understanding and tracking object transformations is critical, and real-time processing is not needed.
>
> **Potential for Amortized Efficiency**: While the spatiotemporal partitioning is presented to be dependent on input object masks for clarity, our actual implementation (available in supplementary code) treats partition construction as a single preprocessing step that is shared across all objects of interest. This design allows the computational cost to be amortized when tracking multiple objects simultaneously. After preprocessing, TubeletGraph operates at speeds comparable to the base SAM2 model.
>
> **Bottleneck Analysis**: The primary bottleneck is in constructing the spatiotemporal partition, which requires independent SAM2 tracking for each identified spatial region. This process is inherently parallelizable. Furthermore,  we envision that future end-to-end systems can track multiple objects without incurring cost linear in the number of objects in the video. While important for practical deployment, we consider this optimization direction orthogonal to our core contributions of tracking and describing object transformations.
>
>
> ---
> ### **W2. Use of a non-open source models for transformation reasoning and description**
>
> We further evaluate semantic correctness when GPT-4.1 is replaced with the open-source Qwen2.5-VL-7B-Instruct. We show comparable performance across both models below, with Qwen marginally underperforming GPT4.1 in terms of action verb description. For details regarding the experimental setup, please refer to *our response to Reviewer jTZM, W1*.
>
> |  |avg. rating|% of 5/5|ICC(2,k)|
> |-|:-:|:-:|:-:|
> |GPT4.1 - verbs|4.28/5 |71.8|0.976 |
> |GPT4.1 - result objs |4.38/5 |71.8|0.923|
> |Qwen 2.5 VL - verbs |3.92/5 |64.1|0.946 |
> |Qwen 2.5 VL - result objs|4.18/5|69.2|0.982 |
>
> ---
> ### **W3. What effects do occlusions have on the model?**
>
> This is indeed a very interesting question. During occlusion events, the model would generally add additional tubelets for the object that is temporarily lost. If the additional tubelet matches the semantic and proximity constraints, it will be incorporated in the tracked object. Since the proximity constraint relies on the three candidate masks predicted by SAM2, the base tracker’s internal candidate masks would set an upperbound for the recall under occlusion. In terms of the transformation, since the VLM only sees the two frames where the initial and resulting objects are visible, it will not be able to describe the occlusion event as a transformation.
>
> Nevertheless, addressing the occlusion challenge with the aid of the VLM is an interesting direction for future works, and we will add discussions on this in the final manuscript.
>
> ---
> ### **W4. Why does ReVOS outperform on M3-VOS?**
>
> Indeed, ReVOS outperforms our approach on M3-VOS, but it significantly underperforms on other datasets with object transformations. According to its authors, they “assume SAM2 takes more training data similar to EPIC-kitchens [5] from which the VOST is constructed.” We share that belief that M3-VOS and DAVIS17 contain far less ego motion compared to VOST. As such, the lower performance of our approach on M3-VOS compared to ReVOS can be traced to the fact that our base SAM2.1 tracker is seeing a distribution shift, compared to the ReVOS tracker.
>
> This also explains the differences on DAVIS17. Given very similar performances for TubeletGraph and base SAM2 models, we show that our method doesn't overfit to transformation scenarios / degrade performance when objects do not undergo transformation.
>
> We will add discussions on this in the final manuscript.
>
> ---
> ### **W5. Limited temporal localization evaluation and confusing choices for varying threshold values**
>
> We acknowledge the concern. Due to the constrained rebuttal period, we are not able to expand the label set further, but we promise to extend the evaluation to the entire VOST val set in the camera ready.
>
> The varying T values were chosen to see how much recall / precision we can gain by setting the proximity threshold to be low (0) and setting the semantic threshold to be relatively high (0.9) respectively. We found that increasing the proximity threshold does not lead to increasing precision, since SAM2’s internal candidate masks, which are what the proximity prior relies on, induce an upper bound. We will incorporate this in the final version.
>
> ---
> ### **W6. Semantic consistency brings little improvement upon ST partition and proximity constraint**
>
> This is indeed a good observation. The semantic constraint does provide small, but consistent gain across all metrics when included. There are cases where semantic cases help, for instance when the tracker latches on to hands or other actors that begin interacting with the tracked object. In these cases, the proximity prior alone is not sufficient to reject the negative candidate. We will provide more example videos / discussion in the final version.
>
> ---
> ### **Q1. Details for fine-tuning SAM2 on VOST**
>
> No, we do not annotate anything further. SAM2.1(ft) is directly finetuned on the existing annotations in VOST. SAM2.1 finetuning does not require the transformations to be explicitly annotated. The finetuned model SAM2.1(ft) tracks the objects in a single binary mask and does not output the transformations (unlike our approach).

---

> > ### Comment · Reviewer_6G8j · 2025-08-02
> >
> > I would like to thank the authors for their detailed answers across all reviews. I find that most of my concerns have been addressed.

---

### Official Review · Reviewer_rGHV · 2025-07-03

**Clarity:** 2
**Significance:** 2
**Originality:** 2
**Rating:** 4
**Confidence:** 4

**Summary:**

This paper introduces TubeletGraph, a zero-shot framework for tracking objects through state transformations while simultaneously detecting and describing these transformations. The authors formulate the "Track Any State" problem, which extends beyond traditional object tracking by maintaining object identity through dramatic appearance changes (e.g., apple being sliced, butterfly emerging from chrysalis). TubeletGraph creates a spatiotemporal partition of the video into "tubelets" using entity segmentation and SAM2 tracking, then uses semantic and spatial proximity constraints to identify missing object components after transformation. The system leverages multi-modal LLMs to generate natural language descriptions of transformations and constructs state graphs representing object evolution. Experimental validation on VOST, VSCOS, M3-VOS, and DAVIS 2017 datasets demonstrates state-of-the-art performance on transformation tracking while maintaining competitive results on standard tracking benchmarks.

**Questions:**

1. The reported 8 seconds per frame processing time on high-end hardware (RTX A6000) severely limits the method's practical applicability. For a tracking method to be useful in real-world scenarios like robotics applications (which the authors cite as motivations), near real-time performance is essential. The lack of thorough complexity analysis or optimization strategies makes this a critical limitation.

​2. The spatial proximity assumption is a fundamental design choice that restricts the method's applicability. The inability to handle non-proximal transformations (e.g., objects that transform while moving significant distances) represents a major limitation that is neither adequately addressed nor tested in the experiments.

**Ethical Concerns:**

["NO or VERY MINOR ethics concerns only"]

**Final Justification:**

Given the responses and the reviews of other reviewers, I will raise my score.

**Limitations:**

yes

**Quality:**

2

**Strengths And Weaknesses:**

Strengths：

1.The paper addresses a genuinely important and challenging problem that extends beyond traditional object tracking. The "Track Any State" formulation is well-motivated with clear real-world applications in robotics and augmented reality.

2.The zero-shot nature of the approach is a significant advantage over supervised methods, eliminating the need for expensive transformation-specific annotations while providing better generalization across different types of object transformations.

Weaknesses：
1. Method validity
​(1) The computational efficiency is a significant concern. The authors report 8 seconds per frame on an RTX A6000 GPU for constructing the spatiotemporal partition, which severely limits real-time applications. While they mention the complexity is near-linear with limited camera motion, this computational cost needs more thorough analysis and potential optimization strategies.
​(2) Limited Handling of Non-Proximal Transformations. The spatial proximity prior (Sec 3.3) assumes transformed objects appear near original locations. This fails for distant transformations (e.g., a launched firework), as evidenced by no such test cases.
2. Experimental details
​(1) The hyperparameter selection process appears somewhat ad-hoc. The authors sweep Tau_prox and Tau_sem on VOST train split and apply the same values across all datasets without further tuning. A more principled approach to hyperparameter selection or adaptive threshold determination would strengthen the method's robustness.
​(2) Method depends on external models (CropFormer, SAM2, FC-CLIP, GPT-4) . The paper would benefit from ablation studies showing the impact of different model choices.

---

> ### Author Rebuttal · Authors · 2025-07-31
>
> We thank reviewer rGHV for their insightful comments and suggestions. We offer details to address the concerns below:
>
>
> ---
> ### **W1. High computational cost**
>
> We acknowledge the concern regarding computational cost and agree that the current speed limits real-time applications. However, we believe TubeletGraph’s unique capabilities provide significant value despite this limitation.
>
> **Value Beyond Real-Time Performance**: TubeletGraph enables detecting and describing object state transformations beyond object tracking in videos. No existing methods perform both temporal localization and semantic description of the transformation in addition to state-of-the-art performance on object tracking under transformation. The combination of tracking and discovering, localizing and describing transformations from videos can be very useful for many applications: for example, producing training annotations on recorded demonstrations for robots,  analyzing compliance and processes on the factory floor from videos, understanding animal behavior and development from camera traps and so on. In these applications, understanding and tracking object transformations is critical, and real-time processing is not needed.
>
> **Potential for Amortized Efficiency**: While the spatiotemporal partitioning is presented to be dependent on input object masks for clarity, our actual implementation (available in supplementary code) treats partition construction as a single preprocessing step that is shared across all objects of interest. This design allows the computational cost to be amortized when tracking multiple objects simultaneously. After preprocessing, TubeletGraph operates at speeds comparable to the base SAM2 model.
>
> **Bottleneck Analysis**: The primary bottleneck is in constructing the spatiotemporal partition, which requires independent SAM2 tracking for each identified spatial region. This process is inherently parallelizable. Furthermore, we envision that future end-to-end systems can track multiple objects without incurring cost linear in the number of objects in the video. While important for practical deployment, we consider this optimization direction orthogonal to our core contributions of tracking and describing object transformations.
>
> ---
> ### **W2. Limited handling of non-proximal transformations**
>
> A launched firework is indeed an interesting example. We believe that this transformation does start with two proximal components (firework leaving the firework box), but due to low frame rate, the two components would appear to be non-proximal. We find these cases to be rare in day-to-day scenarios.
> Furthermore, the proximity constraint in TubeletGraph is soft, as we use the candidate masks produced by SAM2.1 to measure proximity. Therefore, as long as SAM2 can connect these non-proximal objects, our method will still succeed.
>
>
> ---
> ### **W3. The hyperparameter selection process appears ad-hoc**
>
> While we agree that a more principled approach to hyperparameter selection would improve the method's robustness, we found TubeletGraph's performance to be already very robust across varying hyperparameter values. To verify this, we conducted comprehensive grid searches over both spatial proximity and semantic consistency thresholds on M3-VOS and VSCOS datasets. The results demonstrate impressive stability, verifying that the hyperparameters are robust and generalize well across datasets without additional tuning efforts.
>
> | M3-VOS [J]    | prox_thrd=0.1 | 0.2  | 0.3  | 0.4  | 0.5  |
> | -:|:-:|:-:|:-:|:-:|:-:|
> | sem_thrd=0.5 | 74.1  | 74.2 | 74.2 | 74.2 | 74.1 |
> | 0.6  | 74.1| 74.2 | 74.2 | 74.3 | 74.1 |
> | 0.7  | 74.1  | 74.2 | 74.2 | 74.3 | 74.1 |
> | 0.8 | 73.8 | 73.7 | 73.7 | 73.8 | 73.7 |
> | 0.9  | 73 | 72.8 | 72.8 | 72.8 | 72.8 |
>
>
> | VSCOS [J]    | prox_thrd=0.1 | 0.2  | 0.3  | 0.4  | 0.5  |
> | -:|:-:|:-:|:-:|:-:|:-:|
> | sem_thrd=0.5 | 75.6 | 75.8 | 75.9 | 75.8 | 75.8 |
> | 0.6 | 75.6 | 75.8 | 75.9 | 75.8 | 75.8 |
> | 0.7 | 75.7 | 75.8 | 75.9 | 75.8 | 75.8 |
> | 0.8 | 75.6 | 75.7 | 75.8 | 75.8 | 75.8 |
> | 0.9 | 75| 75| 75| 75| 75|
>
> ---
> ### **W4. Missing ablation studies for the impacts of different external model choices**
>
> To address this concern, we conduct ablation studies on key components and their impact on system performance.
>
> **W4.1 FCCLIP vs. DINOv2 Analysis**
> We evaluated our method's robustness by replacing FCCLIP with DINOv2 while keeping all other components unchanged. As shown below, performance remains remarkably consistent across both feature extractors, demonstrating that our framework is robust to the choice of visual encoder without requiring hyperparameter adjustments.
>
> |Method|J|J_S|J_M|J_L|P|R|J(tr)|J(tr)_S|J(tr)_M|J(tr)_L|P(tr)|R(tr)|
> | - |:-:|:-:|:-:|:-:|:-:|:-:|:-:|:-:|:-:|:-:|:-:|:-:|
> |Ours (w/ FCCLIP)|51|41.3|53.3|68.9|68.1|63.8|36.9|23.6|40.5|60.6|55.2|47.2|
> |Ours (w/ DINOv2)|50.9|41.3|52.4|69.4|68.3|63.3|36.7|23.6|39.4|61.3|55.3|46.6|
>
> | Method  | VOST J | VOST J(tr) | VSCOS J | VSCOS J(tr) | M3-VOS J | M3-VOS J(tr) | DAVIS J | DAVIS J(tr) |
> | - |:-:|:-:|:-:|:-:|:-:|:-:|:-:|:-:|
> |Ours (w/ FCCLIP)|51|36.9|75.9|72.2|74.2|64.4|85.6|82.5|
> |Ours (w/ DINOv2)|50.9|36.7|76.2|73|74.1|64.1|85.6|82.5|
>
>
>
> **W4.2 - VLM Robustness for Semantic Evaluation**
> We further evaluate semantic correctness when GPT-4.1 is replaced with the open-source Qwen2.5-VL-7B-Instruct. We show comparable performance across both models, with Qwen marginally underperforming GPT4.1.
>
> |  |avg. rating|% of 5/5|ICC(2,k)|
> |-|:-:|:-:|:-:|
> |GPT4.1 - verbs|4.28/5 |71.8|0.976 |
> |GPT4.1 - result objs |4.38/5 |71.8|0.923|
> |Qwen 2.5 VL - verbs |3.92/5 |64.1|0.946 |
> |Qwen 2.5 VL - result objs|4.18/5|69.2|0.982 |
>
>
> Due to rebuttal period constraints, we will extend this analysis to include CropFormer (replacing with Mask2Former) and SAM2.1 (replacing with Cutie or SAMURAI) in the camera-ready version for a comprehensive robustness evaluation.

---

> > ### Comment · Reviewer_rGHV · 2025-08-03
> >
> > I appreciate the authors’ efforts in addressing the concerns raised. However, considering the feedback from the other reviewers and the current state of the manuscript, substantial revisions are still needed before it can be accepted. Therefore, I am unable to revise my score at this time.

---

> > > ### Author Response · Authors · 2025-08-04
> > >
> > > Thank you for reviewing our rebuttal and providing a timely response. What revisions do you believe are needed before acceptance? We would like to address any remaining concerns that you may have during the discussion period.

---

### Official Review · Reviewer_zLVN · 2025-07-03

**Clarity:** 3
**Significance:** 3
**Originality:** 2
**Rating:** 4
**Confidence:** 3

**Summary:**

The paper introduces TubeletGraph, a new system for tracking real-world objects as they change over time, such as an apple being sliced or a butterfly emerging from a chrysalis. Traditional tracking methods often fail when an object's appearance changes significantly, causing them to lose track of the object. TubeletGraph addresses this by recovering these missing objects and building a state graph that explains what changed and when. It works without any additional training by using spatial and semantic clues to identify relevant parts of the video. It also uses a language model to describe each transforation. The system performs well on multiple datasets, surpassing existing methods in both object tracking and understanding object transformations.

**Questions:**

- Why is SAM2.1++ not considered?
- Can the system handle more complicated or slow changes, not just simple ones like cutting or splitting?
- Is it a problem that the method uses fixed rules to find missing parts of the object?
- How reliable is the description of what happened, since it depends on GPT-4 to explain the changes?

**Ethical Concerns:**

["NO or VERY MINOR ethics concerns only"]

**Final Justification:**

Based on the responses, I now lean towards a borderline accept.

**Limitations:**

Yes

**Paper Formatting Concerns:**

No formatting concerns.

**Quality:**

3

**Strengths And Weaknesses:**

Strenghts:

- The text is well written overall: clear, detailed, and accessible.
- The Paper introduces a task called "Track Any State" that combining object tracking and transformation understanding (to be fair - not sure how new this consideration really is, see e.g.: https://arxiv.org/abs/2404.18143)
- The approach successfully tracks objects even when they change appearance, break apart, or transform over time.
- Recovers missing object parts using a combination of spatial proximity and semantic similarity.
- Builds a state graph that explains what transformation occurred and when it happened.
- Operates in a zero-shot setting without requiring additional training or fine-tuning.
- Achieves improved performance on challenging datasets focused on object transformations.

Weaknesses:

- The progess seems quite incremental (and a bit manually engineered). I don't see a huge methodological advancement to be honest (however, it's not my field of research, hence a lower confidence)
- On the considered datasets, the empirical advantage seems rather small and is only given for 2/4 datasets (performance is not terribly bad on the 2 others)
- The paper does not compare against recent state-of-the-art such as SAM2.1++ (published at CVPR24). At first glance, SAM2.1++ seems to make quite a big step forward. Why do the authors not consider it?

https://openaccess.thecvf.com/content/CVPR2025/papers/Videnovic_A_Distractor-Aware_Memory_for_Visual_Object_Tracking_with_SAM2_CVPR_2025_paper.pdf

- Object recovery depends on fixed thresholds for semantic similarity and spatial proximity (tuned manually).
- While the method is tested somewhat broadly, there is little qualitative or diagnostic analysis of where and why it fails.

Further comments:

- This work seems to align very closely with the ideas of Zack's event segmentation theory. The authors should carefully relate their approach with this well-established theory and related research (also in ML models):

Zacks, J. M., & Swallow, K. M. (2007). Event Segmentation. Current Directions in Psychological Science, 16(2), 80-84. https://doi.org/10.1111/j.1467-8721.2007.00480.x (Original work published 2007)

- Based on the idea of state changes the authors might also have a look at: https://arxiv.org/abs/2305.16291

---

> ### Author Rebuttal · Authors · 2025-07-31
>
> We thank reviewer zLVN for their insightful comments and suggestions. We offer details to address the concerns below:
>
> ---
> ### **W1. Incremental progress**
>
> To clarify, our paper proposes the following contributions:
> - We introduce a novel task of “Track Any State,” which requires simultaneously tracking objects through transformations while detecting and describing those transformations. As shown in Fig. 1 (bottom), no existing methods can both predict complete object tracks and at the same time  provide spatiotemporal grounding and a semantic description of the transformation.
> - We introduce a novel approach for recovering objects after transformations by leveraging a spatiotemporal partition of the input video.
> - Our method tackles the novel task without requiring any labeled transformation data or training, making it broadly applicable across domains.
>
> ---
> ### **W2. Small empirical advantage for 2/4 datasets**
>
> We would like to point out that of the 2 datasets the reviewer is concerned about, DAVIS does not contain transforming objects. We include this benchmark to validate that our method doesn't overfit to transformation scenarios / degrade performance when objects do not undergo transformation. On the other datasets with transformations, our approach provides consistent significant improvements over SAM2.1.
>
> In addition to a consistent, statistically significant improvement over the base SAM2.1 for tracking transforming objects, TubeletGraph does demonstrate an entirely new capability: simultaneous detecting and describing transformations. Here, we would like to note the strong performance in the temporal localization (Sec. 4.2) and semantic accuracy (shown below in *Q3. How reliable is the GTP-4’s description?*).
>
> ---
> ### **W3. Missing comparison with SAM2.1++**
>
> SAM2.1++(CVPR2025) was published after our submission deadline. We benchmarked it via its official repo below.
>
> | Method           | VOST J | VOST J(tr) | VSCOS J | VSCOS J(tr) | M3-VOS J | M3-VOS J(tr) | DAVIS J | DAVIS J(tr) |
> | ---------------- |:-:|:-:|:-:|:-:|:-:|:-:|:-:|:-:|
> |SAM2.1  |48.4    |32.4     |72       |66.9 |71.3|59.3|85.7|83|
> |SAM2.1++|48.8    |33.6     |71.3     |66.0 |72.2|61.3|86.2|84.2|
> |Ours    |51      |36.9     |75.9     |72.2 |74.2|64.4|85.6|82.5|
>
>
> We see that TubeletGraph consistently outperforms SAM2.1++ on all transformation-focuses datasets: VOST (+3.3 J_tr), M3-VOS (+3.1 J_tr), and VSCOS (+6.2 J_tr), while maintaining similar performance on non-transformation dataset: DAVIS (85.6 vs 86.2 J)
>
>
> ---
> ### **W4. Object recovery depends on fixed thresholds**
>
> We found TubeletGraph's performance to be robust across varying threshold values. To this, we conducted comprehensive grid searches over both spatial proximity and semantic consistency thresholds on M3-VOS and VSCOS datasets. The results demonstrate impressive stability, verifying that the hyperparameters are robust and generalize well across datasets without additional tuning efforts.
>
> | M3-VOS [J]    | prox_thrd=0.1 | 0.2  | 0.3  | 0.4  | 0.5  |
> | -:|:-:|:-:|:-:|:-:|:-:|
> | sem_thrd=0.5 | 74.1  | 74.2 | 74.2 | 74.2 | 74.1 |
> | 0.6  | 74.1| 74.2 | 74.2 | 74.3 | 74.1 |
> | 0.7  | 74.1  | 74.2 | 74.2 | 74.3 | 74.1 |
> | 0.8 | 73.8 | 73.7 | 73.7 | 73.8 | 73.7 |
> | 0.9  | 73 | 72.8 | 72.8 | 72.8 | 72.8 |
>
>
> | VSCOS [J]    | prox_thrd=0.1 | 0.2  | 0.3  | 0.4  | 0.5  |
> | -:|:-:|:-:|:-:|:-:|:-:|
> | sem_thrd=0.5 | 75.6 | 75.8 | 75.9 | 75.8 | 75.8 |
> | 0.6 | 75.6 | 75.8 | 75.9 | 75.8 | 75.8 |
> | 0.7 | 75.7 | 75.8 | 75.9 | 75.8 | 75.8 |
> | 0.8 | 75.6 | 75.7 | 75.8 | 75.8 | 75.8 |
> | 0.9 | 75| 75| 75| 75| 75|
>
>
> ---
> ### **W5. Limited qualitative / diagnostic analysis of failure modes**
>
> **Qualitative Analysis**: We showcase representative failure cases in the supplementary HTML page.
>
> *Representative failure cases in transformation descriptions*
>
> Video ID: 0315_paint_cream_4
> - Error: The VLM misidentifies dropped paint cream as "meringue"
> - Analysis: Both share similar visual appearance (white, creamy texture). With a plain background and limited contextual cues, the VLM cannot distinguish between two.
>
> Video ID: 7359_fold_tape_measure
> - Error: The tape measure case is misrecognized as a "smartphone," and the extended measuring tape is described as a "metal rod." Consequently, the transformation is described as "attach" rather than the correct "extend" or "pull out."
> - Analysis: We believe that this failure stems from incomplete object views in the selected frames passed to the VLM, where hand occlusions prevent accurate object recognition and lead to cascading error for the action description.
>
> *Representative failure cases in tracking performance*
>
> Video ID: 0340_fry_omelet_1
> - Error: A false positive omelet is incorporated into the state-graph at t=6s.
> - Analysis: the flipping of a nearby irrelevant omelet causes a new tubelet to initiate, which was falsely incorporated to the original tracked batter due to high semantic and proximity similarity.
>
> Video ID: 0357_make_pancakes_3
> - Error: False negative regions are found during object transformation.
> - Analysis: This is likely caused by a fast rotating occluder spatula altering the appearance of the transforming object rapidly. Once the transformation is complete, all associated regions are properly tracked.
>
> **Diagnostic Analysis**: We observe a reduction in recall as we improve precision via the filterings (Tab. 1). This indicates that some positive objects are lost as we filter out unrelated instances.
>
> We promise to add more failure cases and the above discussion in the camera ready.
>
>
> ---
> ### **W6. Missing references**
>
> We appreciate the references mentioned!
>
> Indeed, the event segmentation discussed by Zacks and Swallow [A] does relate to the predicted state graph well. In fact, when evaluating temporal localization, each annotated transformation does fall near an event boundary, with an altered object state indicating a low-level goal of the activity being achieved. However, not all event boundaries will contain a state-altering transformation (two people playing catch can have well-defined action boundaries without any transformation to the ball). In our task of “Track Any State,” the goal is to detect and describe any object state transformations while maintaining consistent track of the target object.
>
> While Voyager [B] focuses on LLM-driven embodied agent acquiring new skills in a simulated world engine without human intervention, its mechanism of prompting LLM for generating new tasks and programs is similar to our usage of prompting VLM to describe the detected transformation from before/after visual observations (Fig. 2(3)).
>
> The benchmark by Wu et al. [C] focuses on bounding box tracking of transforming objects without any notation of detecting or describing the transformation. In our proposed “Track Any State,” the goal is both detecting and describing any object state transformations and maintaining accurate object tracks.
>
> We will integrate these discussions in the camera-ready version.
>
> [A] Zacks & Swallow. “Event Segmentation.” Current Directions in Psychological Science (2007).
> [B] Wang et al. “Voyager: An open-ended embodied agent with large language models.” arXiv preprint arXiv:2305.16291 (2023).
> [C] Wu et al. “Tracking Transforming Objects: A Benchmark.”  arXiv preprint arXiv:2404.18143 (2024).
>
>
> ---
> ### **Q1. Can the system handle more complicated or slow changes?**
>
> Yes! Please refer to the video examples in the supplementary HTML page. We list some representative examples below.
> - Video ID: 0315_paint_cream_4: dropping multiple droplets of paint cream.
> - Video ID: 0269_press_pencil_1: pressing a pencil tip until it shatters
> - Video ID: 0250_splash_liquid_2: dropping a disk into a bowl of water
> - Video ID: 0369_cut_cheese_3: cutting into a cheese ball containing both solid and liquid components.
> - Video ID: 0357_make_pancakes_3: Spreading pancake batter
> - Video ID: 7359_fold_tape_measure: Extending and folding a metal tap measure
>
> ---
> ### **Q2. Is it a problem that fixed rules are used to find missing object parts?**
>
> Although a more elaborate system may improve performance, we believe that the fixed rules/thresholds allow a very robust system without need for dataset-specific tuning (shown above in *W4. Object recovery depends on fixed thresholds*), at the same time leveraging the modeling capabilities of the underlying models (e.g., FCCLIP for semantic similarity and SAM2’s candidate masks for proximity measure).
>
> ---
> ### **Q3. How reliable is the GTP-4’s description?**
>
> We found TubeletGraph to predict reliable action verbs (4.28/5) and resulting objects (4.38/5) as evaluated by human raters with very high reliability (ICC(2,k) > 0.9). For details regarding the experimental setup, please refer to *our response to Reviewer jTZM, W1*.
>
> |  |avg. rating|% of 5/5|ICC(2,k)|
> |-|:-:|:-:|:-:|
> |Verbs|4.28/5 |71.8|0.976 |
> |Result objects |4.38/5 |71.8|0.923|

---

> > ### Comment · Reviewer_zLVN · 2025-08-07
> > **Concerns addressed**
> >
> > Many thanks for your thorough response. I feel that most of my questions have been addressed. You're right that SAM2.1++ was officially published later (though it appeared on arXiv in late 2024).
> > Overall, I still don't consider the paper a game changer. Nonetheless, based on your responses, I’m willing to slightly raise my scores. I believe the paper could be accepted if it receives support from the other reviewers as well.

---

### Official Review · Reviewer_jTZM · 2025-07-06

**Clarity:** 3
**Significance:** 3
**Originality:** 4
**Rating:** 4
**Confidence:** 4

**Summary:**

While existing object trackers often fail when objects undergo transformations due to significant changes in appearance, this paper introduces TubeletGraph, a zero-shot system designed to recover these lost objects and map out how they evolve over time. The core of the method is to first partition the video into a dense set of potential tracks, or "tubelets," and then intelligently filter them based on spatial proximity and semantic consistency with the original object. For each recovered track, the system uses its appearance as a marker to prompt a multi-modal LLM, which in turn generates a natural language description of the transformation to build a structured state graph. This novel framework allows TubeletGraph to achieve state-of-the-art tracking performance under transformations while also demonstrating a deeper, semantic understanding of the events taking place.

**Questions:**

- The paper states the filtering settings were tuned on the VOST dataset and then applied to the others. Did the authors also try to find the optimal settings for the other datasets, and if so, how much did the performance improve?
- Can you show any examples where the LLM's description of the transformation is incorrect or vague?
- Given that the fine-tuned SAM2.1 (ft) model demonstrates strong performance (Table 1), it would be insightful to see the results of applying the TubeletGraph pipeline on top of this specialized tracker. Did the authors run this experiment? This would help quantify the upper-bound performance of the proposed framework when combined with a domain-adapted tracker.

**Ethical Concerns:**

["NO or VERY MINOR ethics concerns only"]

**Final Justification:**

The authors addressed my main concerns by conducting additional experiments and semantic evaluation.

**Limitations:**

yes

**Paper Formatting Concerns:**

no concerns

**Quality:**

2

**Strengths And Weaknesses:**

Strength:
- The paper defines the important problem of tracking objects undergo transformations, a case where standard trackers often fail.
- TubeletGraph is a training-free method, so it can use the generalization power of the standard trackers like SAM2.1 without needing expensive new training data
- By generating a state graph with descriptions from an LLM, the method provides richer information about what happened to the object, not just where it went.

Weaknesses:
- While the paper provides a quantitative evaluation for the temporal localization of transformations, it lacks a similar evaluation for the semantic correctness of the state graph. The system's ability to "describe the transformation" is only validated through qualitative examples, with no metrics to measure the accuracy of the LLM-generated descriptions
-The paper reports a high computational cost, stating that its method takes "around 8 seconds per frame". This slow speed (roughly 0.125 frames per second) is a major concern for practical usability. While the generated state graph provides rich and useful descriptions of the transformation, as shown in the qualitative examples, this extreme computational demand calls the method's value proposition into question. For many applications, the additional benefit may not justify such a drastic performance drop compared to real-time trackers.
- While the paper does provide an ablation study on its own filtering constraints (semantic and proximity) , it doesn't analyze the impact of the external models it depends on, such as CropFormer, SAM2.1 or CLIP. The paper lacks an analysis of how errors from these foundational models would propagate, which makes it hard to judge the system's overall robustness.
- The paper states that the filtering settings (spatial proximity and the semantic consistency)were tuned on the VOST dataset and then applied to the others. Did the authors also try to find the optimal settings for the other datasets, and if so, how much did the performance improve? the generalization of these selected thresholds was not presented clearly

---

> ### Author Rebuttal · Authors · 2025-07-31
>
> We thank reviewer jTZM for their insightful comments and suggestions. We offer details to address the concerns below:
>
> ---
> ### **W1. Missing semantic evaluation for predicted state graphs**
> To address this, we conducted additional experiments to evaluate the semantic accuracy of the predicted state graphs.
>
> **Experimental Setup**: We first manually annotate captions for each object transformation used in our temporal localization evaluation (Section 4.2). Then, we consider any predicted transformations within 2 seconds of the GT localization as temporally correct candidates for evaluation. To measure semantic accuracy, we consider both the predicted *action verb* and *resulting object state*. We ask 3 human raters to evaluate the semantic similarity between predicted descriptions and ground truth annotations on a 1-5 scale.
>
> |  |avg. rating|% of 5/5|ICC(2,k)|
> |-|:-:|:-:|:-:|
> |Verbs|4.28/5 |71.8|0.976 |
> |Result objects |4.38/5 |71.8|0.923|
>
> **Results**
> - We find high semantic correctness for both the predicted action verbs (average rating of 4.28/5 with 71.8% receiving perfect 5/5 scores) and resulting object states (average rating of 4.38/5 with 71.8% receiving 5/5 scores).
> - The reliability of the human raters are confirmed by intraclass correlation coefficients (ICC) all above 0.9.
> - This validates that the state graphs accurately capture the semantics of the object transformation.
>
> Due to the constrained rebuttal period, we are not able to extend the label set further, but we promise to extend both the semantic and temporal localization evaluation to the entire VOST val in the camera-ready.
>
>
> ---
> ### **W2. High computational cost**
>
> We acknowledge the concern regarding computational cost and agree that the current speed limits real-time applications. However, we believe TubeletGraph’s unique capabilities provide significant value despite this limitation.
>
> **Value Beyond Real-Time Performance**: TubeletGraph enables detecting and describing object state transformations beyond object tracking in videos. No existing methods perform both temporal localization and semantic description of the transformation in addition to state-of-the-art performance on object tracking under transformation. The combination of tracking and discovering, localizing and describing transformations from videos can be very useful for many applications: for example, producing training annotations on recorded demonstrations for robots,  analyzing compliance and processes on the factory floor from videos, understanding animal behavior and development from camera traps and so on. In these applications, understanding and tracking object transformations is critical, and real-time processing is not needed.
>
> **Potential for Amortized Efficiency**: While the spatiotemporal partitioning is presented to be dependent on input object masks for clarity, our actual implementation (available in supplementary code) treats partition construction as a single preprocessing step that is shared across all objects of interest. This design allows the computational cost to be amortized when tracking multiple objects simultaneously. After preprocessing, TubeletGraph operates at speeds comparable to the base SAM2 model.
>
> **Bottleneck Analysis and Potential Improvements**: The primary bottleneck is in constructing the spatiotemporal partition, which requires independent SAM2 tracking for each identified spatial region. This process is inherently parallelizable. Furthermore,  we envision that future end-to-end systems can track multiple objects without incurring cost linear in the number of objects in the video. While important for practical deployment, we consider this optimization direction orthogonal to our core contributions of tracking and describing object transformations.
>
>
> ---
> ### **W3. Missing analysis of external models and error propagation**
>
> To address this concern, we conduct ablation studies on key components and their impact on system performance.
>
> **W3.1 FCCLIP vs. DINOv2 Analysis**
> We evaluated our method's robustness by replacing FCCLIP with DINOv2 while keeping all other components unchanged. As shown below, performance remains remarkably consistent across both feature extractors, demonstrating that our framework is robust to the choice of visual encoder without requiring hyperparameter adjustments.
>
> |Method|J|J_S|J_M|J_L|P|R|J(tr)|J(tr)_S|J(tr)_M|J(tr)_L|P(tr)|R(tr)|
> | - |:-:|:-:|:-:|:-:|:-:|:-:|:-:|:-:|:-:|:-:|:-:|:-:|
> |Ours (w/ FCCLIP)|51|41.3|53.3|68.9|68.1|63.8|36.9|23.6|40.5|60.6|55.2|47.2|
> |Ours (w/ DINOv2)|50.9|41.3|52.4|69.4|68.3|63.3|36.7|23.6|39.4|61.3|55.3|46.6|
>
> | Method  | VOST J | VOST J(tr) | VSCOS J | VSCOS J(tr) | M3-VOS J | M3-VOS J(tr) | DAVIS J | DAVIS J(tr) |
> | - |:-:|:-:|:-:|:-:|:-:|:-:|:-:|:-:|
> |Ours (w/ FCCLIP)|51|36.9|75.9|72.2|74.2|64.4|85.6|82.5|
> |Ours (w/ DINOv2)|50.9|36.7|76.2|73|74.1|64.1|85.6|82.5|
>
>
>
> **W3.2 - VLM Robustness for Semantic Evaluation**
> We further evaluate semantic correctness when GPT-4.1 is replaced with the open-source Qwen2.5-VL-7B-Instruct. We show comparable performance across both models, with Qwen marginally underperforming GPT4.1.
>
> |  |avg. rating|% of 5/5|ICC(2,k)|
> |-|:-:|:-:|:-:|
> |GPT4.1 - verbs|4.28/5 |71.8|0.976 |
> |GPT4.1 - result objs |4.38/5 |71.8|0.923|
> |Qwen 2.5 VL - verbs |3.92/5 |64.1|0.946 |
> |Qwen 2.5 VL - result objs|4.18/5|69.2|0.982 |
>
>
> **W3.3 - Error propagation**
> After analyzing the failure cases, we identified that errors typically manifest as (1) false positive predictions by the base tracker and (2) minor reduction in tracking recall when applying semantic / proximal constraints (Tab. 1). False positive errors (1) can cause erroneous measures of semantic / proximal similarities, while reduction in tracking recall can reduce recall for temporal localization of object transformations.
>
> Due to rebuttal period constraints, we will extend this analysis to include CropFormer (replacing with Mask2Former) and SAM2.1 (replacing with Cutie or SAMURAI) in the camera-ready version for a comprehensive robustness evaluation.
>
> ---
> ### **W4. No threshold tuning on other datasets (generalization is not shown)**
>
> We did not tune hyperparameters on other datasets since we observed TubeletGraph's performance to be robust across varying threshold values. To validate this observation and address the generalization concern, we conducted comprehensive grid searches over both spatial proximity and semantic consistency thresholds on both M3-VOS and VSCOS datasets.
>
> | M3-VOS [J]    | prox_thrd=0.1 | 0.2  | 0.3  | 0.4  | 0.5  |
> | -:|:-:|:-:|:-:|:-:|:-:|
> | sem_thrd=0.5 | 74.1  | 74.2 | 74.2 | 74.2 | 74.1 |
> | 0.6  | 74.1| 74.2 | 74.2 | 74.3 | 74.1 |
> | 0.7  | 74.1  | 74.2 | 74.2 | 74.3 | 74.1 |
> | 0.8 | 73.8 | 73.7 | 73.7 | 73.8 | 73.7 |
> | 0.9  | 73 | 72.8 | 72.8 | 72.8 | 72.8 |
>
>
> | VSCOS [J]    | prox_thrd=0.1 | 0.2  | 0.3  | 0.4  | 0.5  |
> | -:|:-:|:-:|:-:|:-:|:-:|
> | sem_thrd=0.5 | 75.6 | 75.8 | 75.9 | 75.8 | 75.8 |
> | 0.6 | 75.6 | 75.8 | 75.9 | 75.8 | 75.8 |
> | 0.7 | 75.7 | 75.8 | 75.9 | 75.8 | 75.8 |
> | 0.8 | 75.6 | 75.7 | 75.8 | 75.8 | 75.8 |
> | 0.9 | 75| 75| 75| 75| 75|
>
> **Hyperparameter Robustness Analysis**
> - The grid search result shows stable performance across large parameter ranges, with maximum performance differences across a large set of parameter ranges being minimal (M3-VOS: 74.3 vs 72.8; VSCOS: 75.9 vs 75)
> - The parameters tuned on the VOST train (sem_thrd=0.7, prox_thrd=0.3, found in center grid) perform competitively across all datasets.
> - This verifies that the hyperparameter selection generalizes well across datasets without requiring dataset-specific tuning. This robustness further underlines the stability of our filtering mechanism without a need for precise threshold calibration.
>
> ---
> ### **Q1. Examples where the LLM's description of the transformation is incorrect or vague**
>
> Please refer to the supplementary HTML page for the following representative failure cases.
>
> Video ID: 0315_paint_cream_4
> - Error: The VLM misidentifies dropped paint cream as "meringue".
> - Analysis: Both share similar visual appearance (white, creamy texture). With a plain background and limited contextual cues, the VLM cannot distinguish between two.
>
> Video ID: 7359_fold_tape_measure
> - Error: The tape measure case is misrecognized as a "smartphone," and the extended measuring tape is described as a "metal rod." Consequently, the transformation is described as "attach" rather than the correct "extend" or "pull out."
> - Analysis: We believe that this failure stems from incomplete object views in the selected frames passed to the VLM, where hand occlusions prevent accurate object recognition and lead to cascading error for the action description.
>
> We will provide additional failure case examples and analysis in the camera-ready version.
>
> ---
> ### **Q2. Applying TubeletGraph on top of SAM2.1 (ft) and quantifying upper-bound performance**
>
> Thank you for the suggestion. We evaluated TubeletGraph using the fine-tuned SAM2.1 (ft) model as the base tracker below.
>
> |VOST      |J   |J_S |J_M |J_L |P   |R   |J(tr)|J(tr)_S|J(tr)_M|J(tr)_L|P(tr)|R(tr)|
> | - |:-:|:-:|:-:|:-:|:-:|:-:|:-:|:-:|:-:|:-:|:-:|:-:|
> |SAM2.1(ft)|54.4|46.2|53.8|73.1|70.9|65.5|36.4 |25.7 |35.1|61.3 |53.2 |45.4 |
> |Ours |51|41.3|53.3|68.9|68.1|63.8|36.9 |23.6 |40.5 |60.6|55.2 |47.2 |
> |Ours+FT|54.1|45.6|53.8|72.7|69.4|67.7|37.4 |24.8 |38.3 |63.3 |52.5 |49.1 |
>
> We find that the integration of SAM2.1 (ft) with TubeletGraph (Ours+FT) shows modest improvements over the base TubeletGraph. However, the improvements are smaller than expected, given the strong standalone performance of SAM2.1 (ft).
>
> We reason that is because TubeletGraph specifically addresses false-negative predictions by incorporating new candidate tracks lost due to object transformation. Since SAM2.1(ft) is fine-tuned on VOST to minimize these false negatives caused by transformation, the complementary benefits are naturally reduced.

---

> > ### Author Response · Authors · 2025-08-07
> > **We look forward to your feedback**
> >
> > Dear Reviewer,
> >
> > Thank you for your insightful feedback and valuable suggestions. We have taken steps to address your comments in our response. Please let us know if you have any further questions and we are more than happy to address any remaining concerns.
> >
> > Best regards, Authors

---

> > ### Comment · Reviewer_jTZM · 2025-08-08
> >
> > Dear authors,
> > I would first like to thank you for your efforts in addressing the different questions. The evaluation reported for the semantic similarity looks important, and I suggest including it in the camera-ready version. The analysis of the different components in W3 and SAM2.1 are also valuable, especially given the limited rebuttal time.
> > The authors have addressed my main concerns, so I will revise my initial rating.

---

### Note · Authors · 2025-08-12

We thank the reviewers for their time and valuable feedback. We are encouraged that Reviewers jTZM, zLVN, and 6G8j acknowledge that their main concerns are addressed. We will incorporate all new experiments, discussions, and clarifications into the camera-ready version. The main points of our rebuttals are summarized below.

---
### Computational Efficiency
We acknowledge the concern regarding computational cost, however, we believe TubeletGraph’s unique capabilities provide significant value despite this limitation.
- **Value Beyond Real-Time Performance**: TubeletGraph simultaneously tracks objects and detects/describes their transformations - capabilities no existing method offers. This is valuable for applications like labeling for robotic learning, factory compliance analysis and wildlife behavior studies where understanding transformation is critical and real-time speed is not needed.
- **Efficiency Analysis**: The primary bottleneck (constructing the spatiotemporal partition) is inherently parallelizable and can be amortized when tracking multiple objects simultaneously.
---
### State Graph Evaluations
To address reviewers’ concerns, we evaluate and show high semantic correctness as rated by human raters:
- Predicted actions achieved a 4.28/5 average rating (71.8% perfect scores).
- Predicted resulting objects achieved a 4.38/5 average rating (71.8% perfect scores).
---
### Analysis of External Models
Following reviewers’ suggestions, we conduct ablation studies demonstrating robustness to external model choices.
- **FCCLIP vs. DINOv2**: Replacing FCCLIP with DINOv2 shows consistent performance (51.0 vs 50.9 J score on VOST).
- **GPT-4.1 with Qwen2.5-VL**: Replacing GPT-4.1 with Qwen2.5-VL maintains comparable semantic quality according to human raters (4.28 vs 3.92 for verbs and 4.38 vs 4.18 for resulting objects).

We promise to extend this analysis to CropFormer (replacing with Mask2Former) and SAM2.1 (replacing with Cutie or SAMURAI) in the final version.

---
### Generalization of Selected Thresholds
To address the concerns regarding threshold selections, we conduct comprehensive grid searches over both spatial proximity and semantic consistency thresholds on M3-VOS and VSCOS datasets.
- The results show stable performance across large parameter ranges, with a max J score difference of 74.3 vs 72.8 for M3-VOS and 75.9 vs 75 for VSCOS.
- We found threshold selection generalizes well across datasets without requiring dataset-specific tuning.

---

### Decision · Program_Chairs · 2025-09-17

**Decision:**

Accept (poster)

**Comment:**

Reviewers recommend acceptance. The paper addresses the challenging problem of tracking objects undergoing transformations, framing it as the “Track Any State” task. It introduces TubeletGraph: a training-free method that leverages strong pretrained models, and builds a state graph that localizes objects and describes their transformations over time. Reviewers found the problem formulation meaningful, the approach novel, and the paper clearly written with competitive results across multiple benchmarks.

The main concerns relate to computational efficiency—at ~8 seconds per frame, the method is far from real-time and may limit practical use. While this may be a clear bottleneck for practical application of the method, the problem formulation and solution framing remains novel and worth further explorations.

Reviewers also noted the reliance on external models, fixed hyperparameters, and limited quantitative evaluation of semantic correctness. Missing comparisons to some recent baselines (e.g., SAM2.1++) and incomplete runtime/robustness analysis were also raised. These points were addressed properly, as acknowledged by the reviewers.

Overall, despite these limitations, the novelty of the task formulation and the promising results justify acceptance, with revisions to strengthen evaluation and efficiency discussion.